# Numerical Analyses and Optimizations on the Flow in the Nacelle Region of a Wind Turbine

Pascal Weihing[1], Tim Wegmann[1], Thorsten Lutz[1], Ewald Krämer[1], Timo Kühn[2], and Andree Altmikus[2]

[1]Institute of Aerodynamics and Gas Dynamics, University of Stuttgart, Pfaffenwaldring 21, 70563 Stuttgart, Germany
[2]Wobben Research & Development GmbH, Borsigstraße 26, 26607 Aurich, Germany

**Correspondence:** Pascal Weihing (weihing@iag.uni-stuttgart.de)

**Abstract.** The present study investigates the flow dynamics in the hub region of a wind turbine focusing on the influence of the nacelle geometry on the root aerodynamics by means of Reynolds averaged Navier-Stokes simulations with the code FLOWer. The turbine considered is a generic version of the Enercon E44 converter incorporating blades with flatback-profiled root sections. First, a comparison is drawn between an isolated rotor assumption and a setup including the baseline nacelle geometry, in order to elaborate the basic flow features of the blade root. It was found that the nacelle reduces the trailed circulation of the root vortices and improves aerodynamic the efficiency for the inner portion of the rotor, but on the other hand induces a complex vortex system at the juncture to the blade that causes flow separation. The origin of these effects is analyzed in detail. In a second step, the effects of basic geometric parameters describing the nacelle have been analyzed with the purpose to increase the aerodynamic efficiency in the root region. Therefore, three modification categories have been addressed, where the first alters the nacelle diameter, the second varies the blade position relative to the nacelle and the third comprises modifications in the vicinity of the blade-nacelle junction. The impact of the geometrical modifications on the local flow physics are discussed and assessed with respect to aerodynamic performance in the blade root region. It was found that increasing the nacelle diameter deteriorates the root aerodynamics, since the flow separation gets more pronounced. Possible solutions identified to reduce the flow separation are a shift of the blade in direction of the rotation or the installation of a fairing fillet in the junction between the blade and the nacelle.

## 1 Introduction

In recent years, upscaling of wind turbines has led to steadily increasing rotor diameters. Concepts beyond 10MW reveal diameters of more than $200\mathrm{m}$ (Bak et al., 2017). With that development and the particular fact that aero-acoustic emissions and compressibility effects constrain the tip speed to around $80\mathrm{m/s}$, a possible measure to increase the overall rotor power could be an increase of the aerodynamic forces in the inner sections of the rotor. However, structural demands and geometric compatibility are dominant design factors over aerodynamic efficiency, so that typically airfoils with high relative thickness are employed which are blended to cylindrical cross sections towards the hub. The poor aerodynamic performance of these sections involving massive flow separation and complex three dimensional flow structures are well known. It can be shown that by assuming a linear increase of the axial induction towards its optimum value along the radius for the inner one third

of the rotor, $C_P$ cannot exceed a value of $^{43}/_{81}$. This means a loss of $10.5\%$ compared to the Betz value. In practice, the annual energy production losses of conventional turbines due to flow separation in the root region can be estimated to around $3.5\%$ (Loganathan and Gopinath, 2018). In order to increase the rotor efficiency in its inner portion, by an adapted aerodynamic design, a better understanding of underlying flow physics, in particular of the driving effects for flow separation is necessary.

## 1.1 Flow Separation on Conventional Blade Root Geometries and Rotational Effects

As previously mentioned, cylindrical root sections are common due to their structural benefits, cheaper manufacturing and easier transportation. However, regarding their aerodynamic behavior, bluff body like separation is inevitable and even at the airfoils "blending" the root and outer rotor portion strong trailing edge separation is often present as a consequence of angle of attack and in particular thickness induced adverse pressure gradients. Once the flow detaches, the centrifugal force transports the fluid outwards in radial direction. During this motion, the fluid is subject to a Coriolis force accelerating the fluid again in chord-wise direction. These rotational forces are responsible for a delay of stall compared to an equivalent two dimensional section which is commonly known as the Himmelskamp effect (Himmelskamp, 1947). Experimental studies on the rotational effects in the inboard sections have been conducted during NREL's Unsteady Aerodynamics Experiments by Schreck and Robinson (2002) who compared surface pressure data and forces on the rotating blades against analogous stationary conditions. During the MEXICO campaigns (Schepers and Snel, 2007), unsteady surface pressure as well as PIV measurements have been conducted which served as validation basis for studies on stall delay in the hub region by Herráez et al. (2014) and Guntur and Sørensen (2015). At TU Delft, experimental investigations using PIV were performed with focus on the root flow of a two bladed model wind turbine (Akay, 2016). They could identify strong root vortices and high axial velocities in the root region. Recently, Herráez et al. (2016) performed numerical simulations on the same rotor and characterized the Himmelskamp effect as well as the origins of span-wise flow. A similar study has been conducted by Bangga et al. (2018) for the DTU 10MW rotor.

## 1.2 Corner Flow Separation of Aerodynamically Shaped Junctions

The wind turbine considered in the present study differs from conventional designs in the sense that it employs aerodynamically shaped profiles down to the nacelle. This has the benefit that by using suitable flatback airfoils flow separation could be generally eliminated under the isolated consideration of the rotor blade. It could be shown experimentally by Schreck et al. (2013) that for a full scale turbine employing flatback profiles, flow separation was already negligible at $r/R = 0.14$. It must be pointed out that the turbine considered in their study employed a cylinder like connection to the hub, i.e. there was no distinct intersection length with the hub. In case that the airfoil shape is actually maintained towards the root, mutual interaction of the wall boundary layers occurs. Indeed, the melded corner boundary layer is less rich in kinetic energy and therefore prone to so called corner separation. This phenomena is typically found in the junction of the wing and the fuselage of transport aircraft (Levy et al., 2014; Vassberg et al., 2008) or in turbo-machinery cascades (Knezevici et al., 2010). Corner flows are highly influenced by a complex vortex system of primary and secondary vortices and have been investigated in great detail in the experimental work addressing the effect of the horseshoe vortex (HSV) near the leading edge (see review paper of Simpson (2001)) and more recent by Gand et al. (2015), focusing also on the trailing edge and analyzing the corner vortex. The latter

authors found particular evidence that the Reynolds stresses are not aligned with the mean shear tensor, which means that anisotropy of turbulence is present in the region of corner flow separation. Numerically, this makes corner flow separations very challenging to predict. From the NASA drag prediction workshops dedicated to aerodynamic predictions of a transport aircraft (Vassberg et al., 2008) a huge scatter in the separation bubble size of the wing-fuselage junction was obtained in the CFD predictions. Particularly the original version of the Spalart-Allmaras turbulence model (Spalart and Allmaras, 1992) has shown to massively overestimate the bubble size. In order to reduce the model uncertainty for these kinds of flows, NASA very recently initiated an own Junction Flow Experiment (Rumsey et al., 2016).

## 1.3   The Interacting Flow Fields of the Rotor and the Nacelle

For a detailed analyses of the aerodynamic effects in the blade root region the isolated consideration of the the rotor might not be adequate. It is obvious that depending on the shape of the nacelle, displacement effects alter the velocity field acting on the blade. Moreover, the distribution of the bound circulation of the blade might be significantly changed the if there is no air flowing around the root. Previous studies simulating the interacting flow fields of nacelle and the rotor were primarily focused to understand the relationship between the free-stream wind speed and the velocity measured by the nacelle anemometer. For this purpose the simulations in the work of Masson and Smaïli (2006) resolved the detailed geometry of the nacelle and modeled the rotor by an actuator disc approach. In the study of Zahle and Sørensen (2011) the same purpose was pursued but using geometrically fully resolved blades, so that more local effects of the interacting flow fields of the blade and the nacelle could taken into account. The authors showed a non-linear relationship of the local over-speed with wind speed. Regarding the impact of the nacelle on the blade-aerodynamics of the root region, Johansen et al. (2006) redesigned a multi-megawatt rotor by increasing the chord and twist in the root region and further incorporated an egg-shaped nacelle similar to the one considered in the present study. Compared to a conventional rotor with cylindrical sections the thrust and power coefficient could be locally increased for the inner $40\,\%$ of the rotor. The relative improvement of the redesigned blade was found to be higher compared to the additional benefit obtained by including the egg-shaped nacelle.

## 1.4   Improvement of the Aerodynamic Efficiency in the Blade Root Region

A drag reduction of the cylinder like root region of conventional wind turbines blades can only be achieved by diminishing the separated wake flow. Passive or active flow control strategies are promising technologies to favorably influence the boundary layer. Passive solutions to mention are vortex generators  (Baldacchino et al., 2018) that introduce kinetic energy into the boundary, or Gourney flaps and spoilers, that redirect the flow by altering the Kutta-condition. More sophisticated active systems featuring for example tangential blowing (Seifert et al., 1993; McCormick, 2000), or injecting momentum by plasma actuators (Post and Corke, 2004) allow for specific control strategies without generating self-drag or noise of the device. Although these active systems are promising in terms of stall alleviation, the industrial realization on wind turbine blades is still pending, due the enormous technical complexity and high costs.

Despite their structural benefits and improved lift characteristics over conventional thick airfoils, flatback airfoils have the drawback of generating significant drag in companion with blunt trailing edge noise. The reason for this is coherent vortex

shedding that leads to low pressure at the airfoil base. Hence, most of the drag reduction devices aim diminish the interaction of the large scale vortex shedding with the airfoil base. This can be achieved by either breaking up the coherent structures using serrated trailing edges, by hampering the von Kármán instability with splitter plates, or by means of cavities at the trailing edge that shift the shedding vortices away from the airfoil base. An overview on these modifications can be found in Tanner (1975), or Van Dam et al. (2008).

## 1.5  Scope and Objectives

In order to improve the aerodynamic design of turbines in the root region and to stimulate research for such flow control devices described above, a better understanding on the governing aerodynamic effects in the interference region of the blade and the nacelle is extremely important. Particular emphasis in the present study is placed on identifying the driving parameters for root separation with respect to the geometry of the nacelle. In course of this, a modification is introduced in three categories with the purpose to improve the aerodynamic behavior in the root region by diminishing the separation caused by the interfering structures. First, the relative nacelle thickness shall be varied, followed by a variation of the blade position relative to the blade. Moreover, the effect of fillet-type geometry modifications in the blade-nacelle junction shall be discussed. The most promising modifications are examined for off-design conditions, as well.

The next section gives an overview of all considered cases and modifications. In section 3 computational details on the flow solver, settings and grids are provided. The results addressing the above defined objectives are discussed in section 4 and the main conclusions are finally drawn in section 5.

## 2  Baseline Reference Turbine and Considered Modifications

The reference turbine used in this study shall feature the basic geometrical properties of a modern Enercon wind turbine, namely the egg-shaped nacelle and flatback airfoils with large chord in the inboard region of the rotor. For reasons of confidentiality, no original turbine could be used. Instead, a generic re-design of the Enercon E44 converter is employed. The rotor diameter of this pitch regulated turbine is $44$m, tilt and cone angles are set to zero. The induction, as well as the distributions of the airfoil thickness and the solidity are representative for the root region of the industrial counterparts. However, in the generic version the original airfoils have been replaced by open source DU and NACA airfoils. Further, the flatback trailing edge segment is a purely generic design. The latter is responsible for a very high solodity of the rotor in the inner sections as shown in the upper left graph of Fig. 1. The baseline nacelle geometry, is drop-shaped with $L = 8.1$m length and a relative thickness of $48\%$. The point of maximum thickness is at $x/L = 0.39$. The cut position between the rotating hub and the static nacelle is at $x/L = 0.515$.

In order to isolate the governing fluid mechanical effects in the blade root region, different geometrical parameter sets will be addressed which are summarized in Fig. 1. The baseline case described above will be referred to as *CaseT1.0*. The general effects of the nacelle on the blade root flow shall be analyzed in comparison with an isolated rotor simulation (*CaseIsoRotor*). From that, it shall be assessed, whether the isolated rotor assumption holds for rotors with high solidity.

In a second step, the relative thickness of the nacelle is varied from the baseline, being increased by a factor of 1.2 and 1.4, obtaining the modifications *CaseT1.2* and *CaseT1.4*, respectively. For the latter case, the blade has been redesigned in the root region involving a modified twist distribution (*CaseT1.4-twistMod*) which is shown in the upper left graph of Fig. 1. The twist angle has been increased in the root region by around $2°$ and approaches the original distribution at $r/R = 0.36$.

The next modification category addresses a variation of the blade position relative to the nacelle. These modifications are based on the nacelle with the largest thickness. *CaseT1.4-dXm04* denotes a blade shift in negative $x$-direction by $0.4$m, whereas a movement of the blade in rotational direction by $\Delta y = \pm 0.5$m will be investigated in the cases *CaseT1.4-dYm05* and *CaseT1.4-dYp05*, respectively. Additionally, the movement of the blade in direction of rotation shall be applied to the baseline nacelle thickness, as well, yielding *CaseT1.0-dYm05*. The latter incorporates also a hump in the rear part of the suction side of the blade.

Lastly, the geometry in the junction will be modified. In *CaseT1.0-rounded*, the junction line has been rounded by a constant radius of $r = 0.4$m, whereas in *CaseT1.0-fairing*, the radius has been blended from $r = 0.2$m at leading edge and on the pressure side to $r = 0.85$m at the trailing edge of the suction side.

## 3 Computational Details

### 3.1 Flow Solver and Numerical Settings

For the present study, the block-structured finite volume solver of the compressible Navier-Stokes equations FLOWer (Kroll et al., 2000) by DLR has been used. This code is well suited for simulation of rotary wings, since the fluxes caused by relative grid movements are taken into account. Thereby, the movements of the different components are realized with the overset grid technique (Benek et al., 1986). At the authors' institute, FLOWer is continuously developed to improve simulation capabilities of rotary wings, including high order schemes (Kowarsch et al., 2013), advanced turbulence modeling (Weihing et al., 2016), fluid structure interaction (Sayed et al., 2016; Klein et al., 2018), wake modeling (Weihing et al., 2017), and optimization for high performance computing (Letzgus et al., 2018). The code has been extensively validated for wind turbine flows among others during the MexNext projects providing accurate predictions of loads and wake measurements (Schepers et al., 2012; Boorsma et al., 2018). During the EU-AVATAR project very competitive results were obtained in the code-to-code comparisons for both uniform and complex inflow conditions (Sørensen et al., 2014, 2017).

In the present study, the discretization of the Euler fluxes is based on central differences with artificial dissipation according to Jameson et al. (1981) using a $k4$ value of $128$. Unless otherwise stated all simulations are performed steady state. For unsteady simulations, a dual time stepping discretization according to Jameson (1991) is used. Viscous fluxes are discretized by central differences. Regarding turbulence equations, FLOWer uses a fully implicit discretization and has implemented several one- and two-equation models, as well as Reynolds stress closures. Unless otherwise stated all simulations use the $k$-$\omega$-SST model by Menter (1994). As discussed in Appendix A this model is able to predict primary vortices in junction flows and showed good agreement with experiments regarding the extent of corner flow separation.

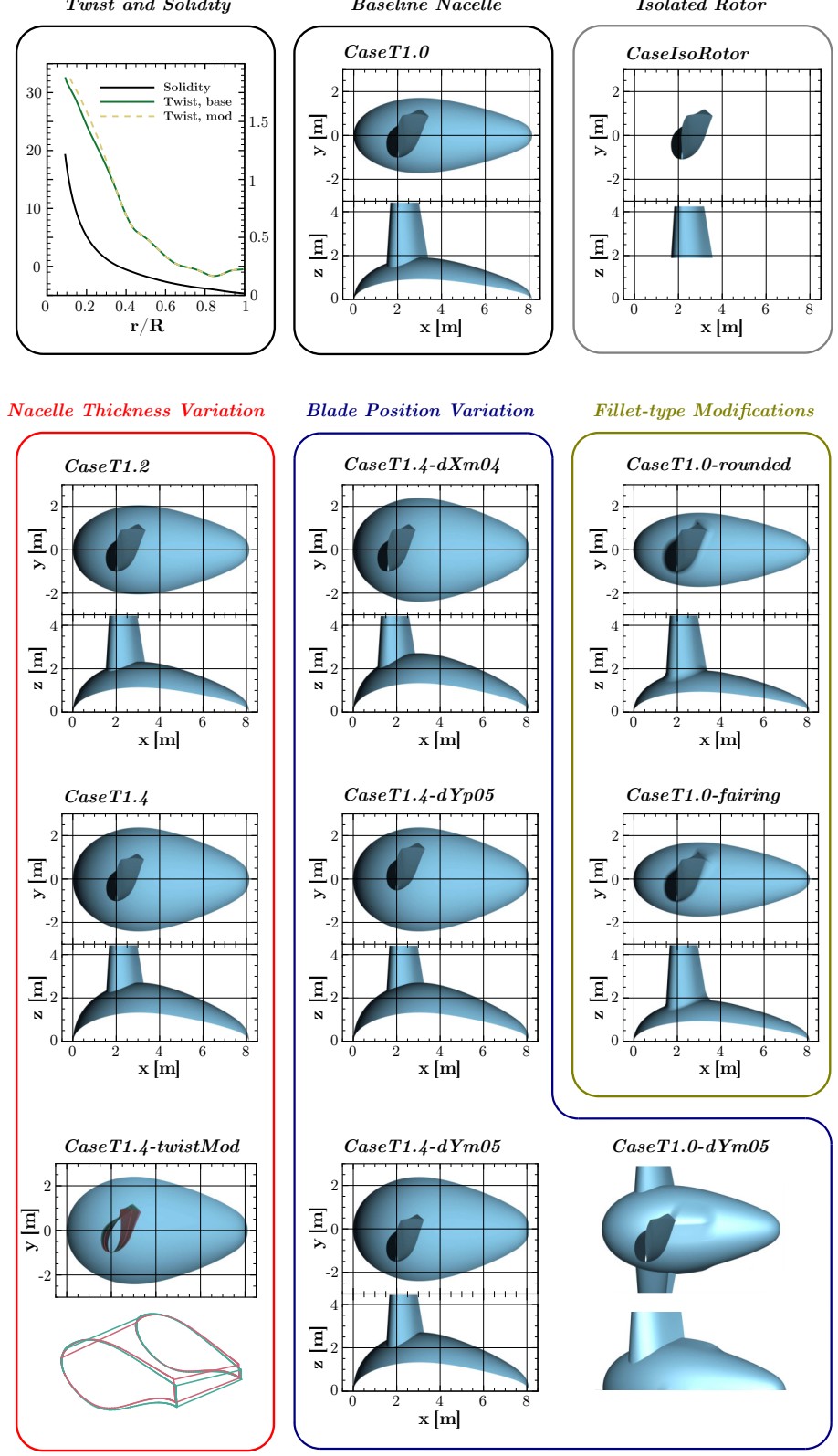

**Figure 1.** Considered geometric variations of the blade-nacelle region.

In order to entirely focus on the impact of the geometric variations on the aerodynamics of the blade-root region and not to bring in additional model uncertainties due to boundary-layer transition, all simulations were conducted fully turbulent. It is pointed out that accurate transition modeling in the considered junction region is non-trivial, since first no engineering model exists for the complex transition mechanism within corner flows and second even outside of the corner-boundary-layer strong cross-flow effects prevail. These are neither taken into account by the basic versions of for example the $e^N$-envelope method (Drela and Giles, 1987) nor the widely used transport equation based local correlation model $\gamma\text{-}Re_\theta$ of Langtry and Menter (2009). To account for cross-flow induced transition it shall be referred to the criteria based model of Arnal and Juillen (1987) or the recent extensions of the $\gamma\text{-}Re_\theta$ model by Grabe and Krumbein (2014); Langtry (2015). Measurements as those of Zamir (1981) further suggest an earlier transition to turbulence within corner boundary layers compared to corresponding flat-plate conditions. Nevertheless, in order to assess the first-order effects of transition on the results, transitional simulations are conducted for the baseline rotor-nacelle configuration by employing the $\gamma\text{-}Re_\theta$ model as well as the envelope method of Drela and Giles (1987). For the latter, the integral boundary layer properties are approximated according to Thwaites (1949). Due to its non-local formulation, requiring a certain grid topology this model was only applied to the structure of the blade.

## 3.2 Grids and Boundary Conditions

The set-up for the simulations considers a periodic $120°$ segment of the flow problem and comprises the meshes for the rotor blade and the nacelle which are embedded into a background grid. For topological reasons, the connections of the blade and the nacelle in *CaseT1.0-rounded* and *CaseT1.0-fairing* use independent grids, which were created manually. All other grids were generated automated by using appropriate *Gridgen* and *Pointwise* scripts, in order to obtain consistent results for all cases.

The polar background mesh extends 24 rotor radii upstream and downstream and 13 radii in radial direction, in order to minimize effects from too close farfield conditions. It contains $448 \times 192 \times 128$ cells in stream-wise, radial and circumferential direction. The grid spacing near the blade is $0.15\text{m} \times 0.3\text{m}$ in the rotor plane.

The mesh of the nacelle covers the whole blade grid and serves as refinement for the root region of the blade. The boundary layer spacing is adapted on the Reynolds number assuring $y_p^+ \approx 1$. The geometric stretching factor is $1.1$ and comprises $64$ extrusion layers. The edge cell size of the latter is equally extended in radial direction up to $r/R = 0.25$ to adequately refine the root region. In circumferential and stream-wise direction $320$ and $168$ cells are used, respectively.

The rotor blade is meshed in a two stage process. All cases use the same blade mesh for $r/R > 0.17$. This part was generated using the *Automesh* script for rotor blade meshing developed at IAG. The chosen dimensions and spacings were based on by the guidelines of the NASA drag prediction workshop Vassberg et al. (2010). The grid has a C-H topology with a H-block extension of the tip using $288 \times 168$ cells in circumferential and radial direction, $64$ cells on the trailing edge and $60$ cells in the wake. Again, a Reynolds number adapted first cell height was chosen and the near wall region is resolved with $64$ layers implying a growth rate of $1.11$. In the second stage the blade mesh is extended in a conformal way to each individual nacelle, including again $64$ cells to refine the boundary layer region of the corner. It was preferred to choose the more expensive H-type refinement of the corner boundary layer in order to avoid skewed cells would that have come when wrapping the blade boundary

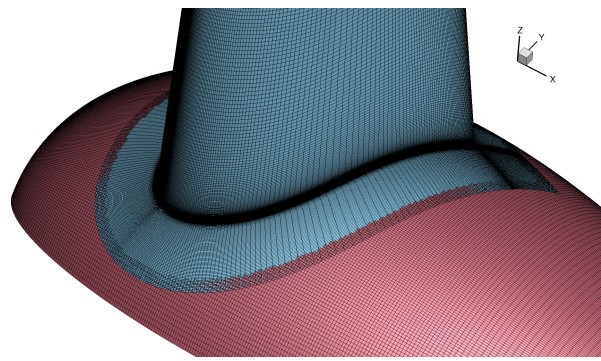

**Figure 2.** Surface grid of blade and nacelle in the corner region.

around the corner. A detailed view of the surface meshes of the blade-nacelle intersection is depicted in Fig. 2 showing the refinement in the junction as well as the overlapping of the grids.

The configurations *CaseT1.0* and *CaseT1.4* have also been also simulated inviscid by the Euler equations. For those simu-
190 lations, the grids have been modified by removing all boundary layer refinements which was necessary to achieve converged solutions.

As mentioned before, the rounding of the junction in the cases *CaseT1.0-rounded* and *CaseT1.0-fairing* does not allow for the previously described grid topology. A high quality grid with smooth cell distributions and as little as possible skewness could only be obtained by manually meshing the blade connector. This was conducted in O-type topology and is shown in
Fig. 3.

The integration of the blade mesh into the nacelle mesh is illustrated in Fig. 4. In total, a typical set-up consists of around 50M grid points.

Regarding the boundary conditions, the 120° domains are mapped periodically and the global domain boundaries carry far-field conditions. All surfaces are treated as no-slip walls (except for the additional Euler simulations of *CaseT1.0* and
200 *CaseT1.4*). However, for the nacelle only the upstream part, the spinner, is rotating (Fig. 4 yellow colored patches), whereas the rear part (brown colored patches) stays fixed. This is achieved in the rotating grid setup without chimera or sliding mesh interfaces under exploitation of periodicity by subtracting the boundary velocity from the value of the interior cell, before setting the ghost layers.

### 3.3 Simulation Parameters and Operating Conditions

The main operating point considered is the one where the power coefficient is maximum, namely at a wind speed of $U_\infty = 10\text{m/s}$. In that case the tip speed ratio is $\lambda = 6$ and the blade pitch angle is zero.Additionally to that point, the configurations *CaseT1.0* and *CaseT1.0-fairing* are analyzed for off-design conditions, as well by considering the further wind speeds, namely $U_\infty = 8; 12; 15\text{m/s}$.

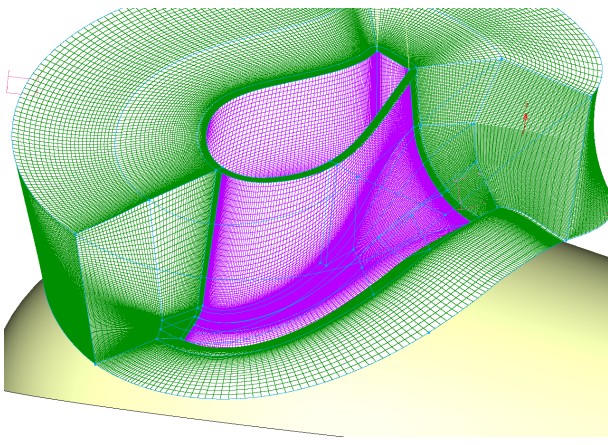

**Figure 3.** Grid topology for the fairing modification.

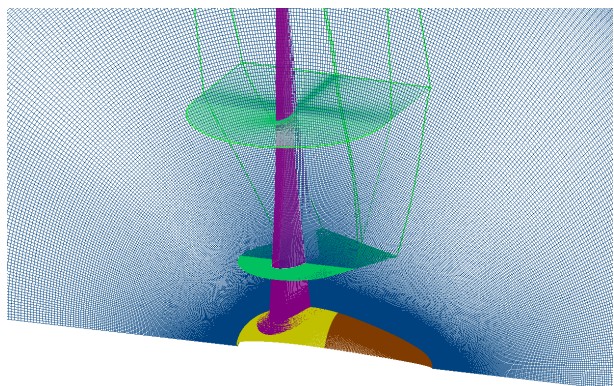

**Figure 4.** Integration of the blade mesh into the nacelle grid.

Unless otherwise stated, all simulations at $U_\infty = 8; 10\,m/s$ have been performed steady state using $150,000$ iterations. To verify convergence, selected simulations have been continued unsteadily by five more revolutions using a time step equivalent to $0.25°$ azimuthal movement. The averaged thrust coefficient of the fifth revolution deviated by less than $0.01\%$ from the averaged value of the last $10,000$ iterations of the steady-state solution. Besides of the integral values, the fluctuations of the radial force distribution in the root region were very small, since only shallow flow separation prevails. For these reason and the fact that computational time can be significantly reduced, all results presented in the following are based on steady-state solutions. At the higher wind speeds $U_\infty = 12; 15\,m/s$ unsteady simulations were performed. Generally, it is clear that for detailed analyses addressing the unsteady behavior of the separated region, unsteady or even scale resolving simulation techniques should be applied.

## 4  Results

In this main section of the paper the simulation results of the different parameter groups shall be discussed, starting with a
comparison of the rotor including the nacelle with the isolated rotor and subsequently analyzing specific geometric variations,
as relative nacelle thickness, blade position and junction shape.

### 4.1  Influence of the Grid Resolution

In order quantify the influence of the grid on the solution, the baseline grid described in section 3.2 shall be compared to a
refined version. The blade grid of the refined setup has the same topology as the one described before. It contains $449 \times 97$
points in circumferential and wall-normal direction, $65$ points on the trailing edge (the base of the flatback) and a refined wake
using $97$ points. In span-wise direction the spacing corresponds to $1\%$ local chord for the inner half of the rotor. The blade-
mesh itself contains $73\,\mathrm{M}$ cells and the total setup incorporating a Cartesian refinement of the near-blade region consists of
$128\,\mathrm{M}$ cells. The radial distributions of the sectional thrust and driving forces are cross-plotted for both resolutions in Fig. 5
and 6. The close coincidence of the forces with almost collapsing curves indicate only a small influence of the baseline grid on
the solution. The integrated thrust force of the medium mesh is $0.139\%$ lower compared to the refined grid. For the integrated
driving force the deviation is even smaller ($0.007\%$), although the local deviation of the sectional loading is slightly larger
compared to the thrust force. Generally, the trends justify the dimensions and spacings for the grid described in section 3.2.
Nevertheless, it will be pointed out in section 4.6 that the effects obtained from the geometric modifications are very local
and their global benefits, e.g. obtained by *CaseT1.0-fairing* at low winds speeds are at the order of accuracy of the used CFD
framework.

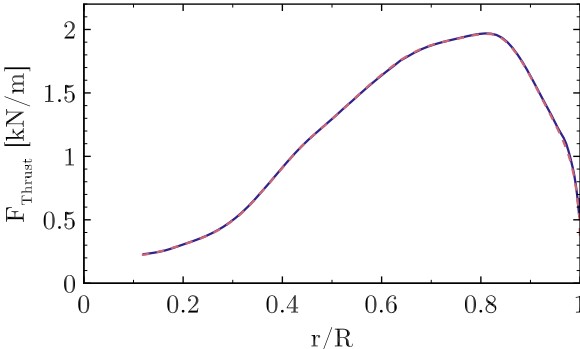

**Figure 5.** Influence of the grid resolution on the sectional thrust force. Standard medium mesh (dashed), refined grid (solid).

### 4.2  General Effects of the Voluminous Nacelle

To investigate the general effects of the nacelle on the blade root aerodynamics, the cases *CaseIsoRotor* and *CaseT1.0* are
compared. The flow field will be characterized showing relevant coherent structures, three-dimensional effects and stall are

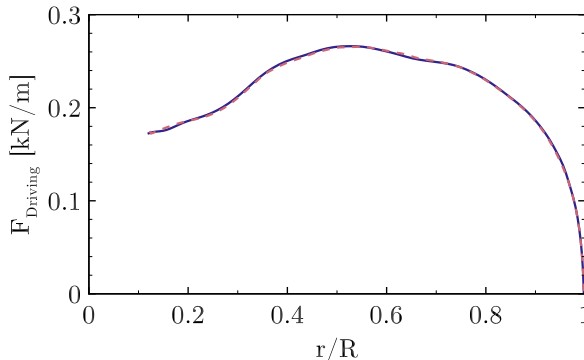

**Figure 6.** Influence of the grid resolution on the sectional driving force. Standard medium mesh (dashed), refined grid (solid).

going to be addressed. A digression will be made on the challenges of determining the AoA in the root region, before the effect of the nacelle on the aerodynamic coefficients will be summarized.

### 4.2.1 Vortical Structures

A first impression of the flow field can be gained by visualizing the dominant vortical structures using the $\Lambda_2$ criterion as well as the axial velocity distribution in the center cut, as shown in Fig. 7. For *CaseIsoRotor* the typical root vortices emerge from balancing the bound circulation at the root. The contour color of the iso-surface denotes the vorticity in $y$-direction and indicates the expected sense of rotation. The normalized axial velocity $u/U_\infty$ in the slice shows a distinct jet through the hub which is responsible for the relatively high advance rates of the root vortices. Apart from those, no other relevant vortical structures are present that might indicate e.g. flow separation. Turning to *CaseT1.0*, shallow flow separation is visible on the suction side near the junction with the nacelle. Moreover, the dominant root vortex seen before has vanished. As will be discussed in more detail in Sec. 4.3, the vortices spiraling around the nacelle evolve from the two vortex legs of the HSV generated in the blade-nacelle junction. As the convecting velocity is significantly smaller than in *CaseIsoRotor*, these vortices which are counter-rotating come very close to each other until eventually, mutual interaction occurs. This results into the formation of three dimensional turbulent structures in the wake of the nacelle. The application of eddy resolving simulation techniques such as DES could bring further insight into these interaction phenomena (Weihing et al., 2016).

An effect that could be identified to enhance the generation of turbulent structures in the nacelle wake is the consideration of both, the rotating spinner and the steady rear part of the nacelle. In previous simulations during the project, where the entire nacelle had been rotated, these structures were almost absent (Kühn et al., 2018). In order to elaborate the effects of the boundary layer roll up and detachment in the rear part of the nacelle, isolated simulations of only the spinner and the static rear part of the nacelle have been conducted. The vorticity in stream-wise direction evaluated in the inertial frame of reference is shown in Fig. 8. The spinner is rotating in positive convention around the $x$-axis and accelerates the surrounding fluid that has no circumferential component far off the wall and therefore generates a negative $x$-vorticity above the spinner. At the interface

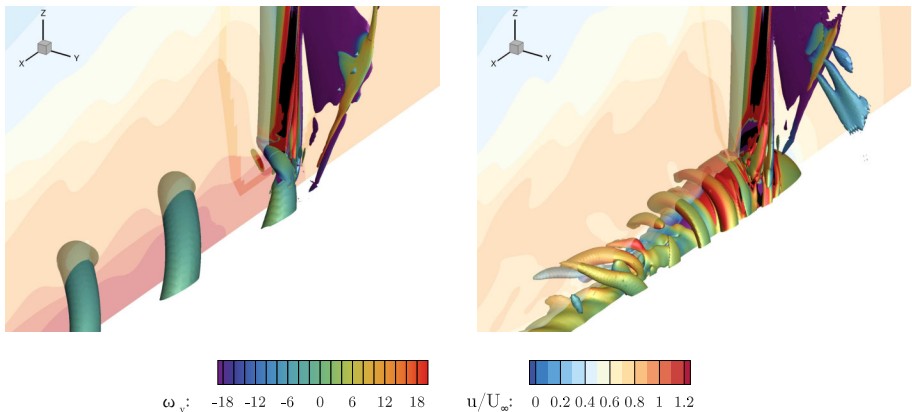

**Figure 7.** Vortices in the root area visualized by a $\Lambda_2$ iso-surface for *CaseIsoRotor* (left) and *CaseT1.0* (right). The vortices are colored by vorticity in $y$-direction. The $x = 0$ slice shows the normalized axial velocity component.

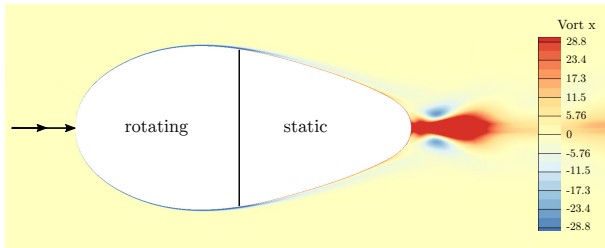

**Figure 8.** Vorticity in stream-wise direction in the boundary layer of an isolated nacelle including only the spinner and the static rear part.

to the non-rotating part of the nacelle this accelerated layer is suddenly retarded again, inducing a positive $x$-vorticity in the near wall region above the static surface. This results into a growing shear layer with positive circumferential velocity that eventually detaches from the surface and forms spiraling longitudinal vortices.

Turning to the mid-portion of the wake, a wavy velocity distribution can be observed for both cases at around $r/R = 0.3$. It is dedicated to trailed vorticity associated with the span-wise gradients of bound circulation. However, its strength appears to be significantly lower compared to the vorticity directly at the root.

### 4.2.2 Three Dimensional Flow and Separation in the Root Region

The extent of the flow separation as well as three dimensional flow patterns can be retrieved from the surface streamlines on the suction side of the blade shown in Fig. 9. For *CaseIsoRotor* there is no flow separation at the root. The slight radial flow component is dictated by the root vortex rolling from the pressure to the suction side. This is also reflected in the velocity contours in a slice $x = 2$m just cutting the leading edge of the blade. The plotted velocity contours indicate the deviation velocity $\tilde{w}$ describing the vertical motion relative to the ideal circular path. Negative values as for *CaseIsoRotor* mean that the streamlines stem from further outboard than assumed from kinematics.

By including the nacelle, its displacement effect evokes an effective inclination angle in that plane which results in a positive deviation component $\tilde{w}$. For the inner 20% of the rotor it is more than 0.5m/s. The negative values at around $r/R = 0.3$ are subject to the induction of the mid-trailing vortex described in the previous section. At the inboard sections, a clear flow separation evolves from the junction with the nacelle. It spreads in radial direction and realigns with chord direction at about $r/R = 0.17$. When focusing on the pressure contours, those are mostly parallel for *CaseIsoRotor* except for the very inboard region that is directly influenced by the root vortex. For *CaseT1.0* significant curvature of the isobars is visible in the area covered by the dividing streamline. As part of the separation process it can be observed that the angle of the shear stress vector turns relative to the pressure gradient, indicating a complex interaction of centrifugal-, Coriolis- and pressure forces as drivers of the three dimensional flow. Due to the very high solidity of the blade, which varies between $c/r \approx 1.1$ at $r/R = 0.1$ and $c/r \approx 0.5$ at $r/R = 0.2$ a high impact of three dimensional effects can be expected.

To estimate the share of these forces acting in the $yz$-plane, the simple balance can be written for a rotation around the $x$-axis as

$$0 = - \begin{pmatrix} \frac{\partial p}{\partial y} \\ \frac{\partial p}{\partial z} \end{pmatrix} + \rho \begin{pmatrix} \Omega^2 y \\ \Omega^2 z \end{pmatrix} + \rho \begin{pmatrix} 2\Omega w \\ -2\Omega v \end{pmatrix}, \tag{1}$$

where the second and third term denote the centrifugal and Coriolis forces, respectively. The velocity components $v$ and $w$ are acting in chord- and span-wise direction relative to the moving blade rotating with $\Omega$. As can be directly seen in the equation, by definition, the Coriolis force vanishes when the velocity is tangential to the rotation, hence it is expected to be small directly above the nacelle. For the balance in span-wise direction the Coriolis force changes sign with respect to the chord-wise velocity component, acting inward for attached flow and outward for separated flow and particularly vanishing near the dividing streamline. In contrast to that, the centrifugal force component in span-wise direction is simply proportional to the $z$-coordinate. Therefore, at the point of separation, the only force to hold the fluid particle on the path of rotation would be a span-wise pressure gradient $dp/dz$. The latter can be derived from the radial pressure distribution plotted in Fig. 10 for different chord-wise sections. At $x_c/c \approx 0.5$ which is close to the separation point, there is virtually no span-wise pressure gradient. Hence, the fluid moves outward solely due to the centrifugal force. It would accelerate towards the tip due to the increasing centrifugal force with $z$, if there was no Coriolis force. As the outward movement is naturally connected with a positive $w$-component it deflects the flow in chord-wise direction and once there is a chord-wise component and a positive $v$-velocity, an additional inward deflection is induced, until eventually, the flow realigns with the chord direction. The transport of separated fluid in outward direction is known as centrifugal pumping (Lindenburg, 2003) and allows for a pressure recovery also in radial direction, where the adverse pressure gradient is smaller compared to the chord-wise direction (see e.g. $x_c/c \approx 0.85$ in Fig. 10). For equivalent two-dimensional conditions, most of the pressure recovery would therefore occur in the turbulent mixing of the airfoil wake. This explains the commonly observed stall delay of a rotating blade compared to equivalent two-dimensional conditions (e.g. (Lindenburg, 2003), (Snel et al., 1993)).

The current investigations supports the studies of Du and Selig (2000), Lindenburg (2003) and recently Herráez et al. (2016) who explained the radial flow with the centrifugal force as well. In this study, the significantly higher blade solidity caused by the flatback airfoils with large chord is seen to be responsible for a significant impact of three dimensional effects.

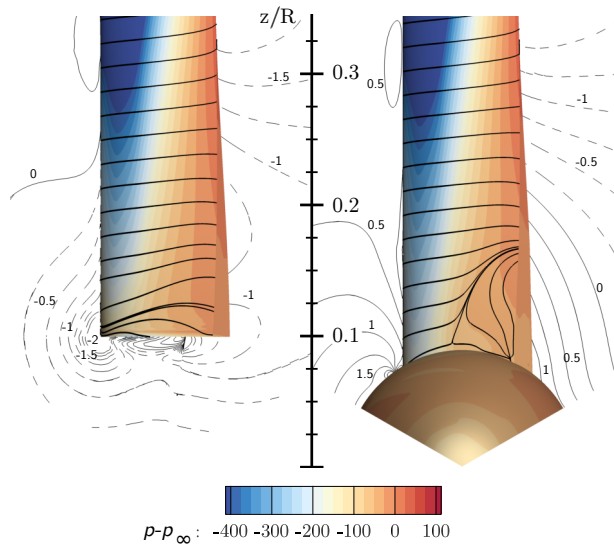

**Figure 9.** Visualization of three dimensional flow in the root region using streamlines and pressure contours on the suction side of the blade for *CaseIsoRotor* (left) and *CaseT1.0* (right). The contour lines indicate the "deviation velocity" $\tilde{w}$ from to the ideal kinematic value $-\Omega y$ in a slice at $x = 2$m.

Turning to the front part of the blade, it can be shown that following Bernoulli's principle, the static pressure decreases in radial direction due to the increasing dynamic pressure of the inflow. This results in an outward pressure gradient. However, it is not believed to be the only reason for the slight radial component of the streamlines near the leading edge, since first, the pressure gradient is balanced by the Coriolis force in negative $z$-direction and further, a strong acceleration in chord-wise direction prevails, which is in accordance with Schreck and Robinson (2002). As described in more detail in Sec. 4.4 it is more related to an effective sweep angle of the blade relative to the inflow direction. This is again an effect of the high solidity of the blade which results into a leading edge shift relative the rotational axis in $y$-direction. Comparing the pressure levels, the suction peak is significantly reduced in *CaseIsoRotor* due the induction of the root vortex over the whole inner portion of the rotor.

### 4.2.3 Determination of the Angle of Attack in the Root Region

In order to assess the aerodynamic performance of the individual blade sections in terms of aerodynamic coefficients and to utilize that information in two-dimensional approaches such as BEM or for airfoil design tools, it is essential to accurately determine the angle of attack (AoA) from the three-dimensional flow field. However, it is clear that its extraction is far from trivial as wind velocity and rotation are superimposed by the induction of the wake and the bound circulation of the blade itself. Most approaches such as those compared by Rahimi et al. (2017) aim to eliminate the effect of the bound circulation. They provide very similar results at the mid-board blade sections. However, at the blade tip and root, major differences were observed. For the root region those were explained by the massive flow separation in the wake of the very thick airfoils. As for

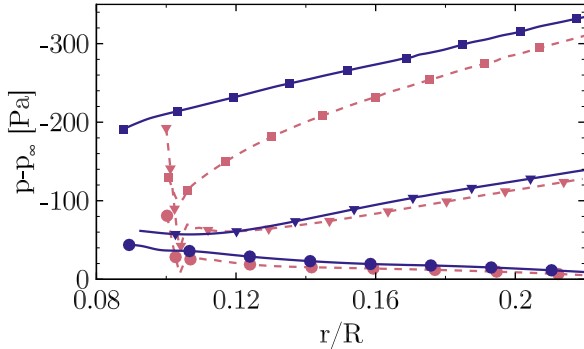

**Figure 10.** The effect of the nacelle on the radial pressure distribution on the suction side, at $x_c/c \approx 0.1$ (squares), $x_c/c \approx 0.5$ (triangles) and $x_c/c \approx 0.85$ (circles) for *CaseIsoRotor* (dashed), *CaseT1.0* (solid)

the present rotor flow separation is only shallow, or even absent, differences between the AoA extraction methods are strongly linked to the effects of the root vortex system, instead of the dominant vortex shedding caused by massive flow separation. Therefore, two AoA evaluation methods shall be compared with focus on the root region.

The first is the *Reduced Axial Velocity* method (RAV) of Johansen and Sørensen (2004), which eliminates the bound circulation by azimuthally averaging the axial velocity for each radial section. Secondly, the *Line Average* method of Jost et al. (2018) is applied which averages the velocity vector along closed lines around the blade. Both methods are sketched in Fig. 11. As seen in the figure, for the *Line Average* method, the extraction of the velocities occurs along circles that are curved about the rotational axis with a radius corresponding to the location of the quarter-chord point in each section, being centered around the latter. The radius of the projected circle is one chord length. By averaging the planar velocity vector along these curves, Jost et al. (2018) could show that the effect of bound circulation can be eliminated resulting in the pure inflow conditions for that section. The evaluation of the AoA in Fig. 12 shows a lower AoA near the tip for the *Line Average* method, since it locally accounts for the induction of the tip vortex, which is smeared out by the azimuthal averaging in the RAV method (c.f. Fig. 11). Accounting for the local induction is a desired feature of the *Line Average* method and has shown to accurately reconstruct the inflow conditions for corresponding two-dimensional analyses of the near tip region (Jost et al. (2018) and Rahimi et al. (2017)). Near the blade root, the AoA predicted by the *Line Average* method is higher than obtained by the RAV method. The reason for this is that the azimuthal arc length for averaging the axial velocity in the RAV method is now smaller compared to averaging over the circular line of the *Line Average* method (c.f. Fig. 11). Particularly, the acceleration of the curving stream-lines upstream of the airfoil are not considered in the RAV method. This effect becomes particularly important for *CaseT1.0*, where both the blade and the nacelle displace the flow and therefore accelerate it upstream of the leading edge which is the reason why the increase of AoA is more pronounced for the case including the nacelle. Additionally, it must be noted that for the RAV method the root vortex is occupying about $40\%$ of the downstream arc length, being more concentrated for the *Line Average* method. As a consequence the latter predicts a higher axial velocity in the rotor plane compared to the RAV method

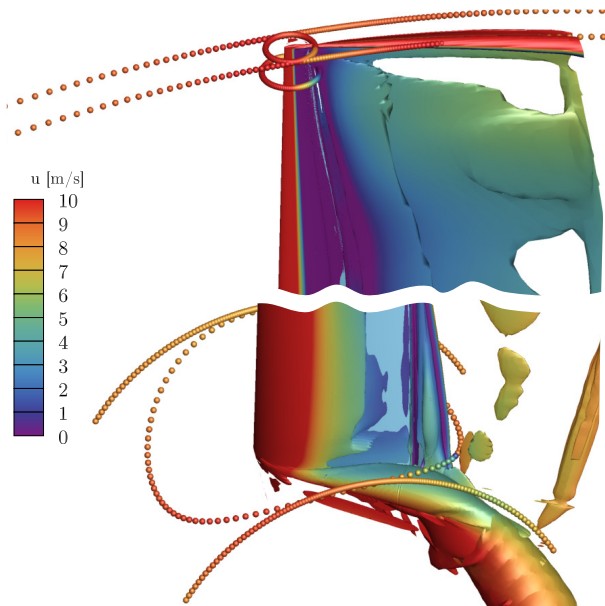

**Figure 11.** Extraction of axial velocity in the inertial frame near the tip and the root using the *Reduced Axial Velocity* method (Johansen and Sørensen, 2004) (circular arcs) and the *Line Average* (closed curves) (Jost et al., 2018) method.

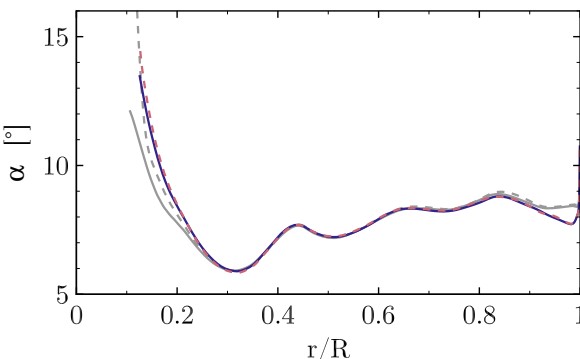

**Figure 12.** Radial distribution of the angle of attack for *CaseIsoRotor* (dashed) and *CaseT1.0* (solid) using the RAV method (gray) and the *Line Average* method (colored).

(c.f. Fig. 13), and therefore a higher AoA. In the remainder of the paper, the AoA will be evaluated with the *Line Average* method.

### 4.2.4 The Effect of the Nacelle on Aerodynamic Coefficients

The pressure coefficient $C_p$ is compared for the configurations *CaseIsoRotor* and *CaseT1.0* as well as two-dimensional airfoil simulations in a slice at $r/R = 0.136$ (Fig. 14). Following Bernoulli's principle, the pressure coefficient has been normalized

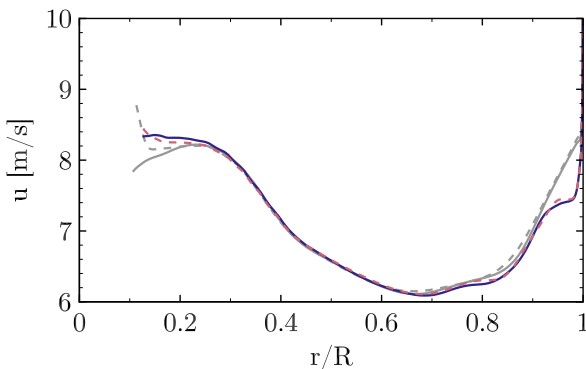

**Figure 13.** Radial distribution of the axial inflow velocity for *CaseIsoRotor* (dashed) and *CaseT1.0* (solid) using the RAV method (gray) and the *Line Average* method (colored).

using the maximum dynamic pressure that can be exploited from the kinematics $q_{\infty,rot} = \rho/2 \left( U_\infty^2 + (\Omega r)^2 \right)$. In accordance with the independence principle of swept wing theory, only the component normal to the leading edge is relevant for the aerodynamic properties. Therefore, the $z$-coordinate has been used as radius for the circumferential component in the dynamic

pressure. It should be noted that one could also account for the induction and sweep angle effects, but then the reference state becomes rather ambiguous and difficult to compare among the different cases. Hence, it will be accepted that with the kinematic normalization, $C_p$ might differ from a value of 1 in the stagnation point and it will be attempted to explain this with the dominant physical effects.

As already discussed in the previous section, the induction of the root vortex significantly diminishes the suction peak for

*CaseIsoRotor*. For the rotor including the nacelle, the pressure level is lower for the entire suction side of the blade, particularly in the front part of the airfoil. Hence higher lift and lower pressure drag can be expected for *CaseT1.0*. At the pressure side the differences are small. Only directly at the stagnation point a higher pressure coefficient is obtained by *CaseIsoRotor*, which is slightly $> 1$. This can be explained by the negative $\tilde{w}$ contours (Fig. 9) which indicate that the fluid hitting the airfoil is actually stemming from further outboard compared to its ideal kinematic path. Therefore, fluid of higher momentum is transported

towards the leading edge part of the airfoil, yielding a higher stagnation pressure than assumed for normalization. By contrast, in *CaseT1.0* the stagnation pressure is lower than estimated, due to an effective inclination angle which pushes fluid from the inboard sections to the reference cut. Finally, focusing on the aft chord region, it can be confirmed that also for *CaseT1.0* the cut is outside of recirculation and that pressure has almost recovered by the centrifugal pumping mechanism.

To further assess the three-dimensionality of the flow in the hub region, a comparison shall be drawn with two-dimensional

simulations. Those were performed at the angles of attack $\alpha = 10.2°$, $13°$ and $14°$, where the first corresponds to the extracted value from the RAV method for *CaseT1.0*, the second corresponds to a slightly higher value than obtained from the *Line Average* method which was $12.3°$ for *CaseT1.0* and the last is close to the AoA obtained for increased nacelle thicknesses as shown in Sec. 4.3. The 2D results at $\alpha = 10.2°$ reveal no separation. At $\alpha = 13°$ shallow trailing edge separation is present which moves forward to $x_c/c \approx 0.6$ at $\alpha = 14°$. Compared to the 3D results, *CaseIsoRotor* shows a completely different behavior

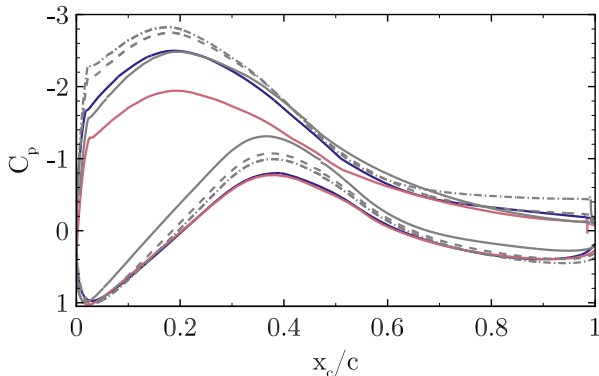

**Figure 14.** Pressure coefficient $C_p$ at $r/R = 0.136$ for *CaseIsoRotor* (solid red) and *CaseT1.0* (solid blue) compared with two-dimensional simulations (gray lines): $\alpha = 10.2°$ (solid), $\alpha = 13°$ (dashed), $\alpha = 14°$ (dashed dot).

which cannot be mimicked by any 2D simulation. At a first glance the level of the suction peak of *CaseT1.0*, suggests that $\alpha = 10.2°$ would be a good approximation. However, it must be pointed out that when scaling the stagnation pressure in the 3D results to a value of 1, the distribution increases a bit. Moreover, the visual inspection of corresponding streamline plots compared in that section (not shown here) clearly suggested a larger AoA than $10°$. The distribution in the adverse pressure regime is very similar to the 2D results at $\alpha = 13°$ implying a similar boundary layer stress. The differences on the pressure

side are also markedly, showing distinctly stronger suction peaks predicted by any of the two-dimensional simulations. These examinations emphasize the strong three-dimensionality of the flow in the root region of the present rotor and point out that the application of two dimensional polars in BEM without correction models must be treated with caution, even if there is no massive flow separation.

    The radial distribution of the circulation $\Gamma$ plotted in Fig. 15 a), as well as the lift coefficient (Fig. 15 b)) confirm the

improvement of the aerodynamics in the root region by including the nacelle which effectively diminishes the harmful inductive effect of the root vortex. This is also reflected in a reduction of the drag coefficient (Fig. 15 c)) by up to 100 drag counts, although *CaseT1.0* involves flow separation. The region of influence of the nacelle extends to $r/R = 0.35$. By integrating the driving force within that range, this results in a higher torque of around $20\%$ for that portion.

### 4.3    The Impact of the Relative Nacelle Thickness

From the previous section it was concluded that an improvement of the aerodynamic properties in the inner portion of the rotor could be obtained by taking into account the nacelle. The attenuation of the root vortex by the nacelle diminished induced drag and increased lift. In this section the effects of the relative nacelle thickness will be analyzed. The thickness has been increased in two steps by a factor of $1.2$ and $1.4$ (*CaseT1.2*, *CaseT1.4*), respectively.

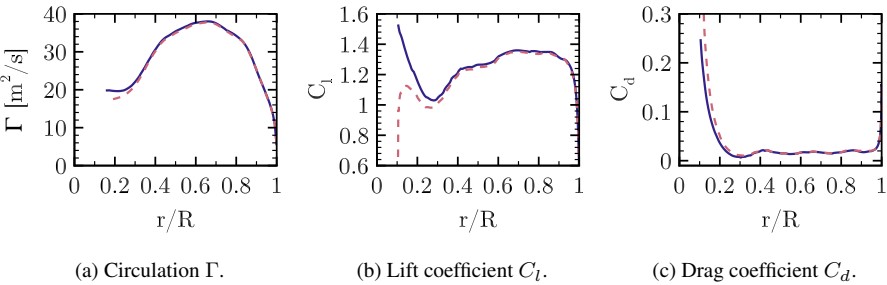

(a) Circulation $\Gamma$.    (b) Lift coefficient $C_l$.    (c) Drag coefficient $C_d$.

**Figure 15.** Distribution of aerodynamic coefficients along the blade radius for *CaseIsoRotor* (dashed) and *CaseT1.0* (solid).

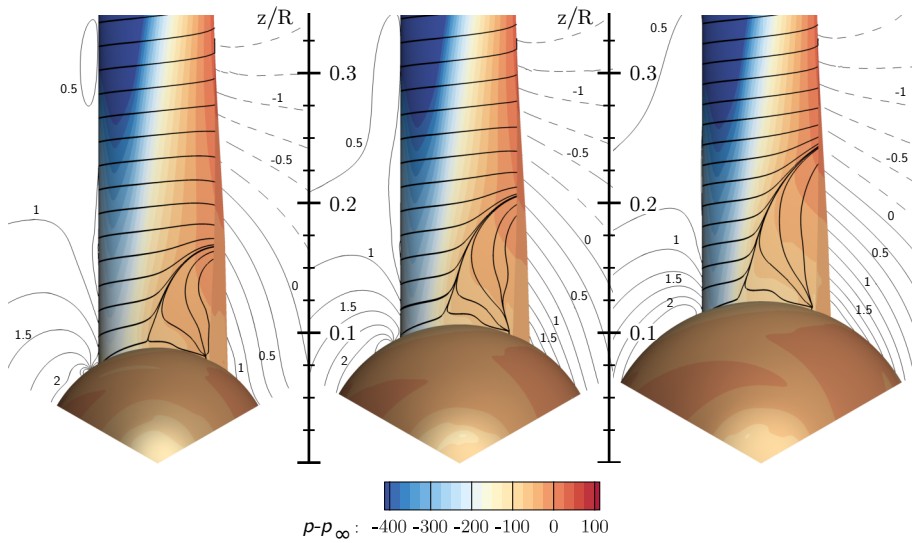

**Figure 16.** Visualization of three dimensional flow in the root region using streamlines and pressure contours for *CaseT1.0* (left), *CaseT1.2* (mid) and *CaseT1.4* (right). The contour lines are plotted for a slice at $x = 2$m and indicate the velocity $\tilde{w} = w - \Omega y$ which denotes the deviation from the ideal kinematic path.

### 4.3.1 The Effect on Flow Separation and its Driving Parameters

The resulting surface streamlines and pressure contours shown in Fig. 16 indicate growing flow separation with increasing nacelle thickness. The extent of the separation measured from the point of maximum thickness of the nacelle to the radial position of reattachment increases from $0.08R$ over $0.10R$ to $0.12R$ for the configurations *CaseT1.0*, *CaseT1.2* and *CaseT1.4*, respectively. With increasing thickness of the nacelle the separation line moves forward by around $15\%$. In addition, it can be noted from the "deviation velocity" $\tilde{w}$ that the inclination of the inflow increases with larger relative nacelle thickness.

The comparison of the pressure coefficient in the reference cut at $r/R = 0.136$ (Fig. 17) confirms the increasing separation for larger nacelle thicknesses. The distinct pressure plateau already suggests a loss of aerodynamic efficiency due to flow

separation. The radial pressure gradients in the front part of the blade (Fig. 21) show that greater relative nacelle thickness increases the suction force due to the displacement effect of the nacelle. Also, in the mid- and aft- chord region the initial pressure level is lower, which increases the radial adverse pressure gradient. Since separation inherently alters the pressure distribution, it is difficult to distinguish between the causes and effects of the flow separation. In order to assess whether stronger adverse pressure gradients develop by increasing the nacelle thickness, inviscid reference simulations have been performed for *CaseT1.0* and *CaseT1.4*. The interaction of the inviscid pressure fields is visualized in Fig. 19 using vectors of static pressure acting on the surfaces. For *CaseT1.0* the flow accelerates moderately in the junction, reaching the minimum of pressure at about $x_c/c \approx 0.5\text{-}0.6$. It can be shown that the adverse pressure gradient imposed by the nacelle is slightly shifted behind the one from the blade. Turning to *CaseT1.4* this is not the case anymore, as the suction peaks fairly coincide at around $x_c/c \approx 0.3\text{-}0.4$. Downstream of that point the adverse pressure gradient markedly increases compared to *CaseT1.0* and is accordingly devolved on the blade as shown in Fig. 17. In *CaseT1.0* the inviscid and viscous suction peaks closely coincide, whereas a significantly higher peak prevails for the inviscid simulation of *CaseT1.4*. This clearly suggests that the boundary layer of the associated viscous case is increasingly loaded with larger nacelle thickness. From these investigations it can be concluded that a reduction of separation might be achieved by decoupling the interfering pressure gradients of the nacelle and the blade as it is typically conducted for winglets.

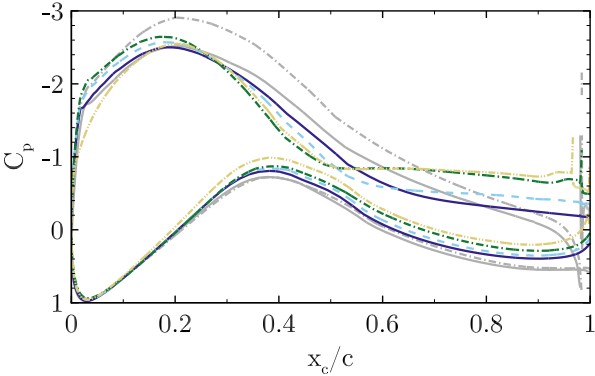

**Figure 17.** Airfoil pressure distribution at $r/R = 0.136$ for *CaseT1.0* (solid), *CaseT1.2* (dashed), *CaseT1.4* (dashed dot) and *CaseT1.4-twistmod* (dashed dot dot). The gray curves denote corresponding inviscid simulations.

A second aspect, that turned out to be crucial for the development of the flow separation is directly linked to the vortex system evolving in the junction of the blade and the nacelle. The motivation for a detailed look into that came across, by analyzing AoA behavior with respect to the different nacelle geometries (Fig. 22), where an increasing AoA of around two degrees could be observed from *CaseT1.0* to *CaseT1.4*. In the beginning of the investigations, it was assumed that the increasing separation was the primary effect of the larger AoA. Therefore, it was attempted to re-design the blade in order to compensate the AoA leading to *CaseT1.4-twistMod* which employs an increased twist angle in the blade root region of around $2.0°$ (see Fig. 1).Except for the sections very closely to the root, the AoA could be effectively reduced. Very inboard, the aerodynamic

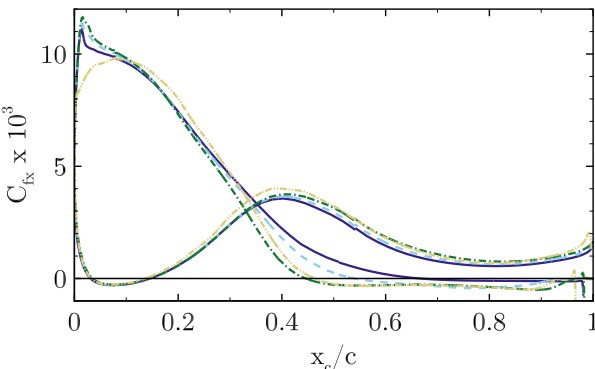

**Figure 18.** Airfoil skin friction distribution at $r/R = 0.136$ for for *CaseT1.0* (solid), *CaseT1.2* (dashed), *CaseT1.4* (dashed dot) and *CaseT1.4-twistmod* (dashed dot dot).

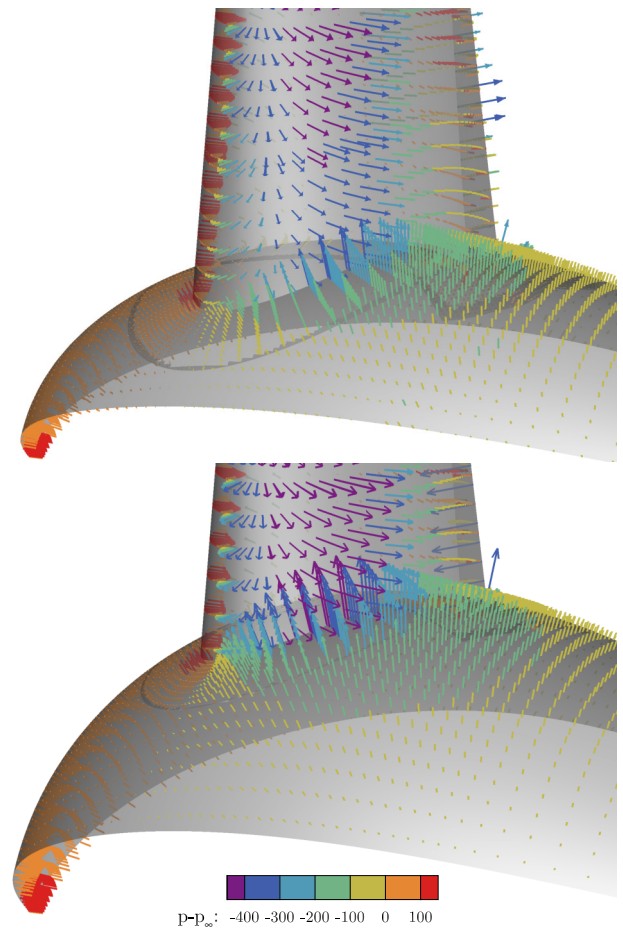

**Figure 19.** Inviscid pressure distribution in the junction region of blade and nacelle for *CaseT1.0* (upper) and *CaseT1.4* (lower).

behavior is obviously strongly non-linear. Due to the twist modification the flow is redirected which causes an increase in the axial velocity. The latter is detected by the *Line Average* method which evaluates a lower AoA reduction than expected by the geometric twist modifications. In turn the RAV method was closer to the kinematic value. Nevertheless, when looking at the pressure distribution in Fig. 17, the suction peak is significantly reduced. The evaluation of chord-wise skin friction (Fig. 18) shows that the separation remains almost the same as for *CaseT1.4*, so it could be concluded that flow separation is not directly affected by an AoA induced adverse pressure gradient on the blade.

It is more seen to depend on the interacting boundary layers in the junction region of the blade and the nacelle. To shed more light into that, the emerging vortices in the junction are visualized in Fig. 20 for *CaseT1.0* and *CaseT1.4* using volume streamlines colored by vorticity in the local direction of the velocity vector. Additionally, volume cuts are placed normal to the blade at the leading edge and at $x_c/c = [0.25; 0.5; 0.75]$, respectively. The horseshoe vortex (HSV) is clearly visible and seems to be generated in the stagnation region, when the boundary layer of the nacelle approaches the blade. It is rolling inward and its size and strength could be observed to increase with larger nacelle thickness. This behavior is consistent with Simpson (2001), who reports a stronger HSV with increasing AoA. Gand et al. (2015) showed experimentally that the onset of corner separation is delayed by a stronger HSV, since fluid of higher momentum is pushed into the blade boundary layer. This beneficial inductive effect, likewise depends on the distance to the blade. By increasing AoA the suction side leg departs from the blade and is further deflected by the Coriolis force.

Directly inboard of the HSV, the counter rotating corner vortex (CV) evolves from the stagnation point and closely follows the juncture of the blade and the nacelle. Its production depends on the velocity gradients of the interacting boundary layers and its strength was observed to increase for the larger relative nacelle thickness in the adverse pressure gradient region. In contrast to the HSV, the CV remains aligned with the junction. Due to its sense of rotation it seems to be responsible to deform the near wall velocity profile on the blade and to pull the boundary layer flow away from the wall. For all cases it was observed that the recirculation area was initiated from the streamline originating in the CV. Thus, high vorticity in the CV might be an important driver for the whole separation process. In particular it should be noted that the configuration with modified blade twist (*CaseT1.4-twistMod*) revealed quasi identical values for the CV strength as *CaseT1.4* and showed the same amount of separation.

Hence, the second strategy to diminish the detrimental flow separation could be a a relief of the mutual loading of the boundary layers of the blade and the nacelle, in order to influence the CV strength and propagation. This aspect shall be investigated in Sec. 4.5.

### 4.3.2 The Effect of the Relative Nacelle Thickness on Aerodynamic Coefficients

To summarize the effects of the relative nacelle thickness on the aerodynamic coefficients in the root region which are plotted in Fig. 22, those confirm the degradation of aerodynamic performance due to stronger flow separation in the inboard region, where lift mostly decreases and particularly drag increases. The AoA increases with the nacelle thickness, since reduced lift decreases axial induction, which yields higher axial velocities in the rotor plane. The decrease of aerodynamic efficiency is most prominent for the inner 25% of the rotor radius. It should be noted that there was no measurable benefit for the outer rotor

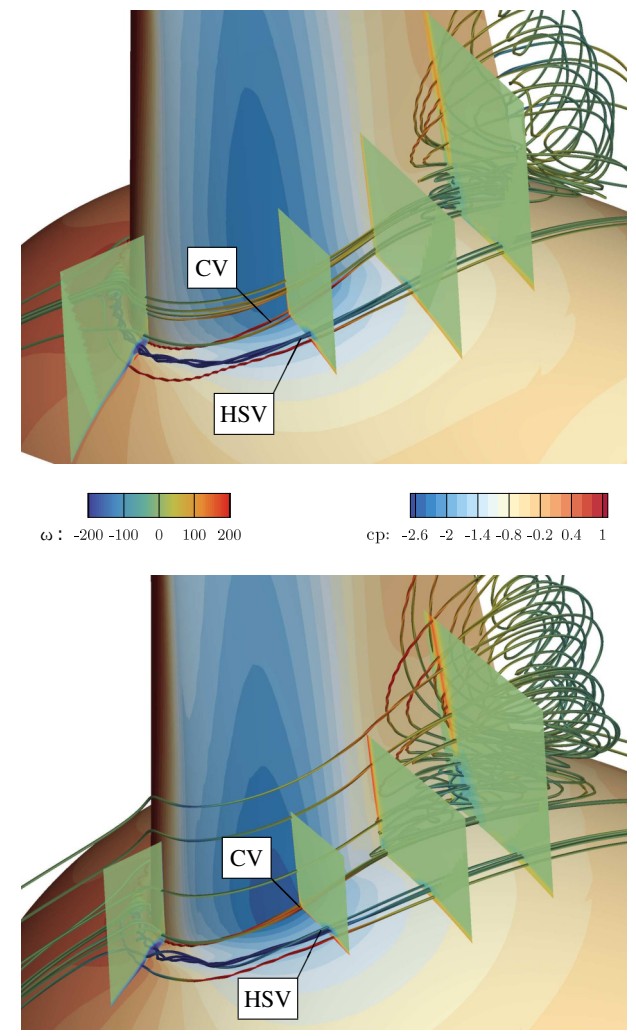

**Figure 20.** Vortex system in the junction of blade and nacelle for *CaseT1.0* (upper) and *CaseT1.4* (lower).

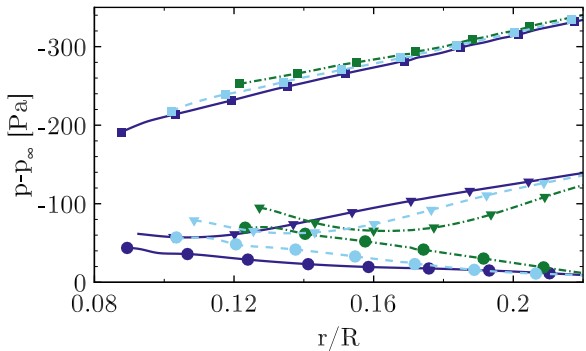

**Figure 21.** The effect of the nacelle thickness on the radial pressure distribution on the suction side, at $x_c/c \approx 0.1$ (squares), $x_c/c \approx 0.5$ (triangles) and $x_c/c \approx 0.85$ (circles) for *CaseIsoRotor* (dashed), *CaseT1.0* (solid), *CaseT1.4* (dashed dot).

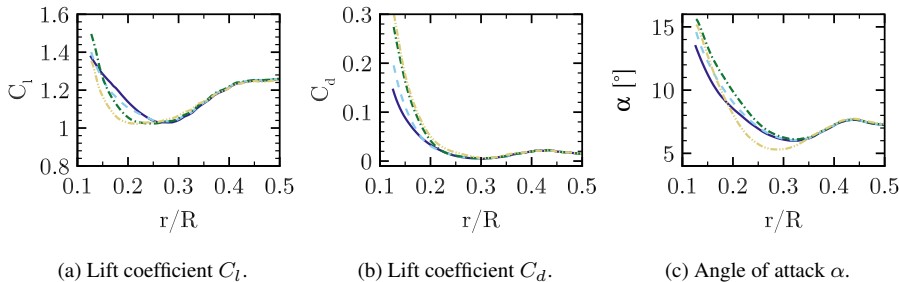

(a) Lift coefficient $C_l$.      (b) Lift coefficient $C_d$.      (c) Angle of attack $\alpha$.

**Figure 22.** Aerodynamic coefficients along the blade radius for *CaseT1.0* (solid), *CaseT1.2* (dashed), *CaseT1.4* (dashed dot) and *CaseT1.4-twistmod* (dashed dot dot).

sections as one might have expected due to a displacement effect of the nacelle. In total, the torque generated by one blade decreased for *CaseT1.4* by $1.18\%$ compared to *CaseT1.0*.

### 4.4 Movement of the Blade Position Relative to the Nacelle

As pointed out in section 4.3.1, a segregation of the interacting pressure gradients of the blade and the nacelle could reduce the overall loading on the corner boundary layer. In order to analyze these effects, the blade was shifted upstream in axial direction (*CaseT1.4-dXm04*), as well as upstream- and downstream in circumferential direction (*CaseT1.4-dYm05* and *CaseT1.4-dYp05*, respectively). All modifications were based on the nacelle with the largest of the considered thicknesses, since first, the strongest effects on the flow separation might be present there, and further, this configuration provides more space for relative blade shifts. The motivation for the first modification was to place the entire blade in front of the point of maximum thickness of the nacelle and, therefore, locate it in the favorable pressure gradient in axial direction. The shifting of the blade in $y$-direction correspondingly aims to investigate the relative location of the pressure gradients in circumferential direction. Bangga (2018)

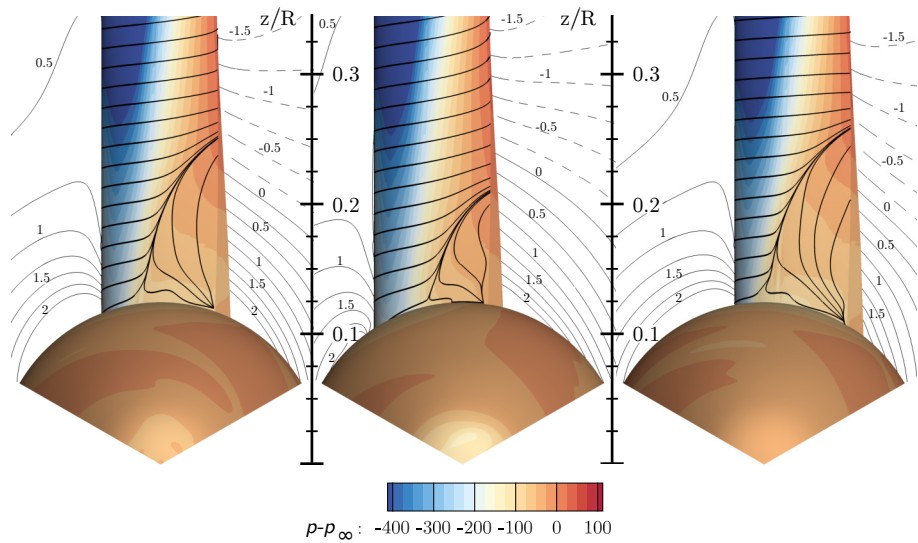

**Figure 23.** Visualization of three dimensional flow in the root region using streamlines and pressure contours for *CaseT1.4-dXm04* (left), *CaseT1.4-dYm05* (mid) and *CaseT1.4-dYp05* (right). The contour lines are plotted for a slice at $x = 2\text{m}$ ($x = 1.5\text{m}$ for *CaseT1.4-dXm04*) and indicate the velocity $\tilde{w} = w - \Omega y$ which denotes the deviation from the ideal kinematic path.

investigated the lateral shift of rotating profiles for the first time and focused on the exploitation of the $y$-component of the centrifugal force in Eq. 1 to increase rotor performance.

### 4.4.1 The Effect of the Relative Blade Position on Flow Separation

Comparing the surface streamlines plotted in Fig. 23 with those of the centered version (Fig. 16), the movement of the blade forward in axial direction reveals no improvement regarding the extension of the corner separation. Indeed, even a slight deterioration is present, which might be caused by the fact that the inclination angle relative to the blade increases. Turning to *CaseT1.4-dYp05*, an increase of the separation extent by around $8\%$ can be observed compared to *CaseT1.4*. Here, separation already begins close behind $30\%$ of chord. A significant improvement can be achieved by shifting the blade in direction of rotation. The separation size in radial direction is $0.086R$, which is only slightly above the baseline *CaseT1.0* but already smaller than in *CaseT1.2*.

The boundary layer profiles in the reference cut $r/R = 0.136$ (Fig.24) give information about the mass flow rate of the recirculation. At $x_c/c = 0.5$, *CaseT1.4-dYm05* is still attached, whereas slight and moderate back flow is observed in *CaseT1.4* and *CaseT1.4-dYp05*, respectively. Turning to the profiles at $x_c/c = 0.75$, the height of separation massively increases when shifting the blade in positive $y$-direction. As seen in Fig. 23 this seems to be linked with the volume covered by the downward curvature after the point of maximum thickness of the nacelle and might be "felt" by the blade boundary layer as an additional expansion, which goes along with added adverse pressure loading. This effect is weaker for *CaseT1.4-dYm05*, where most of the pressure recovery is achieved forward of that point.

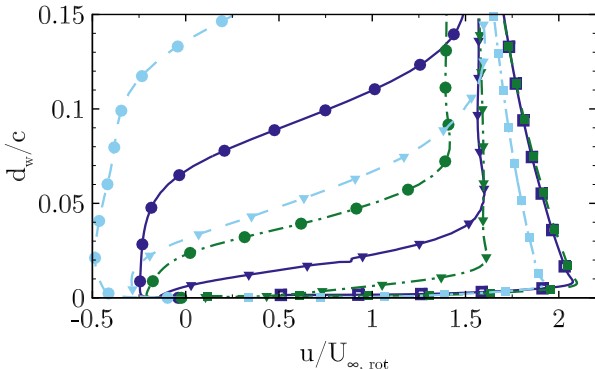

**Figure 24.** Stream-wise normalized velocity in the boundary layer $u/U_{\infty,rot}$ at $r/R = 0.136$ and $x_c/c = [0.25; 0.5; 0.75]$ (squares; triangles; circles), for *CaseT1.4* (solid), *CaseT1.4-dYm05* (dashed dot) and *CaseT1.4-dYp05* (dashed).

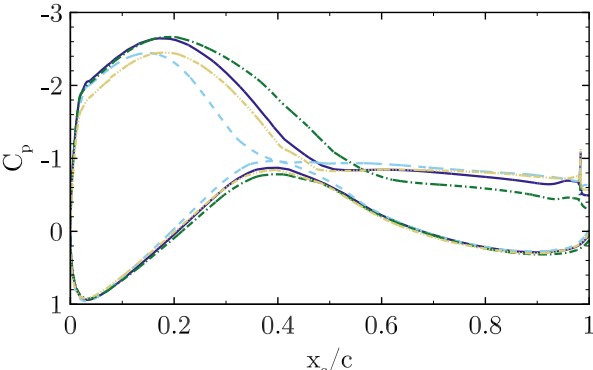

**Figure 25.** Airfoil pressure distribution at $r/R = 0.136$ for for *CaseT1.4* (solid) *CaseT1.4-dXm04* (dashed dot dot), *CaseT1.4-dYm05* (dashed dot) and *CaseT1.4-dYp05* (dashed).

In a similar way, the chord-wise (Fig. 25) pressure distributions are affected in accordance to the previous observations. In *CaseT1.4-dXm04*, the suction peak decreases, but the kink to the pressure plateau remains at the same position as for *CaseT1.4*. In the aft-chord region the pressure recovery has still not initiated as it might be "blocked" by the displacement effect of the ascending nacelle diameter in the vicinity of the blade suction side. In *CaseT1.4-dYp05*, the massive flow separation yields a collapse of lift at $x_c/c \approx 0.4$, which results in a reduced suction peak. Both cause a tremendous increase of pressure drag. The latter can certainly be reduced for *CaseT1.4-dYm05*, where moderate separation begins at $x_c/c \approx 0.65$. The downward slope shows that the last part of the pressure recovery already occurs along the airfoil and not predominantly in the wake, as for the other cases.

Regarding the span-wise pressure distribution (Fig. 26), a slight increase of suction force can be noted near the leading edge in *CaseT1.4-dYm05* for the whole inner portion, whereas it is markedly lower when moving the blade in the other direction. In contrast to the other cases, *CaseT1.4-dYm05* maintains the slope of the leading edge cut, also along $x_c/c = 0.5$. Particularly,

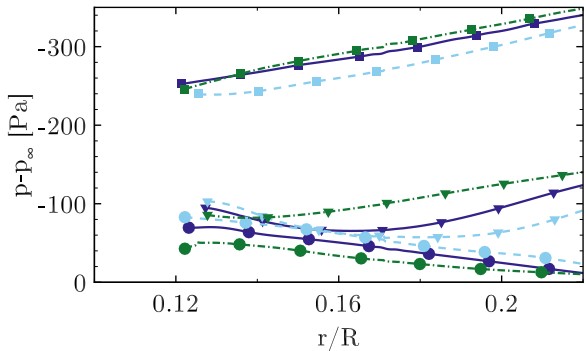

**Figure 26.** The effect of the blade position on the radial pressure distribution on the suction side, at $x_c/c \approx 0.1$ (squares), $x_c/c \approx 0.5$ (triangles) and $x_c/c \approx 0.85$ (circles) for *CaseT1.4* (solid), *CaseT1.4-dYp05* (dashed), *CaseT1.4-dYm05* (dashed dot).

*CaseT1.4-dYp05* already shows the behavior typically found near the trailing edge, where pressure recovery is redeployed in radial direction by the centrifugal pumping mechanism. Consistently at $x_c/c = 0.85$, the radial pressure gradient is smallest in *CaseT1.4-dYm05*, since less recirculating mass needs to be transported outward, which directly yields an earlier realignment of the streamlines in chord-wise direction as seen in Fig. 23.

From the previous investigations in Sec. 4.3.1 using the Euler simulations, it became clear, that a decoupling of the adverse pressure gradients seems necessary to relief the boundary layer in the junction. As could be shown the axial shifting did not yield any improvement, so that it can be deduced that the predominant pressure gradient is the one in lateral direction. This seems also reasonable as the flow in the junction is aligned with that direction. As the adverse pressure gradient imposed by the nacelle initiates at the outmost point in circumferential direction, it follows that the entire pressure recovery is additionally loaded in *CaseT1.4-dY05*. Opposed to that, the segregation of the adverse pressure gradients seems to work for *CaseT1.4-dYm05*.

Another important aspect to be considered is already attributed to the inflow. As shown by the left sketch in Fig. 27, the inner sections of the blade which is shifted forward are affected by inflow stemming from an effectively larger radius that consequently brings in greater momentum. Along the airfoil the flow is pushed downward compared to its ideal kinematic path, which means that it is effectively streaming from larger to smaller radius. According to conservation of angular momentum this results in acceleration, which is supportive to overcome the adverse pressure gradient. When crossing the $xz$-plane a corresponding retardation would prevail, which additionally reduces angular velocity in *CaseT1.4-dYp05*. To illustrate this inflow hypothesis, a velocity difference plot between *CaseT1.4-dYm05* and *CaseT1.4* is presented in Fig. 27 for the reference cut $r/R = 0.136$. For this visualization, the solution of *CaseT1.4-dYm05* was mapped on top of the centered version. For the entire front part of the airfoil a higher velocity magnitude prevails, which is particularly present inside the boundary layer as seen before in Fig. 24. In the stagnation region the inflow velocity is $\approx 1\mathrm{m/s}$ higher for *CaseT1.4-dYm05* compared to *CaseT1.4*. Another distinct peak, where the velocity is markedly higher for *CaseT1.4-dYm05* is found in the region of adverse

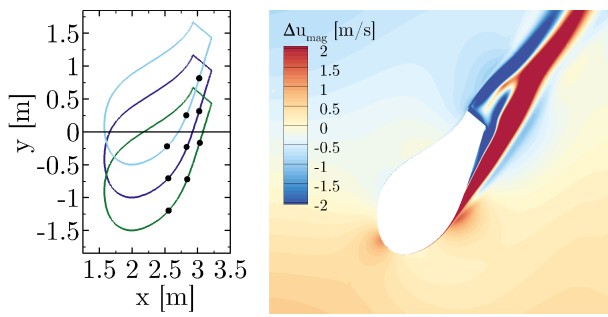

**Figure 27.** Sketch of the shifted profiles at $r/R = 0.136$ (left). Difference in velocity magnitude between *CaseT1.4-dYm05* and *CaseT1.4* (right).

pressure at around $x_c/c \approx 0.5$. In the rear part of the airfoil the differences originate from the different thickness of the separated wake.

The third effect that is seen to be beneficial for the delay of separation by shifting the blade in direction of rotation is the sweep angle of the inflow vector with respect to the blade leading edge. At the reference cut it comprises more than $25°$ for *CaseT1.4-dYm05*, about $17°$ for *CaseT1.4* and around $9°$ for *CaseT1.4-dYp05*. In radial direction the sweep angle decreases exponentially, but is still greater than five degrees at the tip for *CaseT1.4-dYm05*. As the flow turns about the airfoil, the effective sweep angle is not constant along the chord-wise direction, but since most of the lift is generated in the front portion of the blade, it can certainly be stated that sweep effects cannot be neglected. This is supported by the streamlines in *CaseT1.4-dYm05* (Fig. 23) that show the typical pattern found for swept wings at high AoA (Obert, 2009), featuring a distinct attachment line instead of a stagnation point, which results in $C_{p,max} < 1$, as well as a prominent span-wise deflection of the streamlines near the trailing edge. In contrast to that, the streamlines remain approximately perpendicular to the leading edge in *CaseT1.4-dYp05*. The question, whether the principle of independence holds or not is certainly debatable, since rotation introduces span-wise gradients to the flow field. Indeed, for attached boundary layers it is a typical assumption made in BEM codes and is also supported by analyses of Leishman (1989). At high AoA in the stall regime it is generally not valid anymore. However, as observed in the boundary layer profiles (Fig. 24), the streamlines (Fig. 23) and pressure distribution (Fig. 25), there is evidence that the sweep angle delays stall and is beneficial for reattachment as it stimulates the outward centrifugal pumping mechanism. An overview of the effect of sweep angle on dynamic stall can be found in the text book of Leishman (2006). Measurements reinforcing the present observations regarding the effect of sweep angle on maximum lift coefficient and stall delay can be found for example in Dwyer and Aiccroskey (1971), or in Purser and Spearman (1951).

### 4.4.2 The Effect of the Blade Position Relative to the Nacelle on Aerodynamic Coefficients

Regarding the global consequences of the relative blade position, those shall be compared in terms of AoA, lift- and drag coefficients plotted for the inner rotor half in Fig. 28. The AoA seems generally to increase when flow separation becomes stronger which was already seen for the cases where the nacelle thickness had been increased. Compared to *CaseT1.4*, the

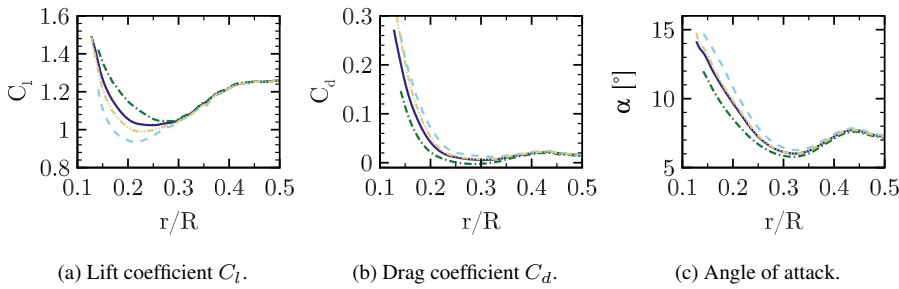

(a) Lift coefficient $C_l$.  (b) Drag coefficient $C_d$.  (c) Angle of attack.

**Figure 28.** Aerodynamic properties along the blade radius for *CaseT1.4* (solid), *CaseT1.4-dXm04* (dashed dot dot), *CaseT1.4-dYm05* (dashed dot) and *CaseT1.4-dYp05* (dashed).

axial movement of the blade shows to decrease torque by $0.96\%$ as lift declines and drag slightly rises. The significantly stronger flow separation in *CaseT1.4dYp05* further deteriorates aerodynamic efficiency and is reflected in a decline of torque by even $1.98\%$. *CaseT1.4dYm05* clearly increases lift and decreases drag in the inner portion of the rotor which raises torque by $2.1\%$. This configuration is already better than *CaseT1.2*, but still around $1.5\%$ worse than *CaseT1.0*. However, as will be shown in the next section, a movement of the blade in direction of rotation based on *CaseT1.0* is able to outperform the aerodynamic behavior of the latter.

### 4.5  Fillet-Type Modifications in the Junction

The second strategy for a potential reduction of corner separation elaborated in Sec. 4.3.1 was the relief of the boundary layer interaction of blade and nacelle which shall be addressed in this section. The considered configurations modify the geometry of the junction by applying a constant radius of $0.4$m all around the airfoil, denoted *CaseT1.0-rounded* and introducing a fairing on the suction side of the blade referred to as *CaseT1.0-fairing*. These modifications are based on *CaseT1.0*. Since the movement of the blade in direction of rotation showed promising results when being based on the largest nacelle, this modification shall be transferred to the smallest nacelle (*CaseT1.0-dYm05*) and be included for comparison in the present discussion. In the rear part the junction reveals a plateau which, from another case not shown here (Kühn et al., 2018), turned out to be beneficial regarding a restriction of the recirculation area. Since the lateral shift on this nacelle brings along conflicts regarding meshing when moving the blade forward, the periodic segment had to be rotated by $17°$, but due to axisymmetry, this does not change anything regarding periodicity.

### 4.5.1  The Effect on Flow Separation and the Corner Vortex System

The streamline plot in Fig. 29 shows that flow separation can be entirely suppressed for *CaseT1.0-dYm05* and *CaseT1.0-fairing*. For the latter quasi two-dimensional flow conditions prevail with only slight streamline deflection close to the trailing edge, whereas the shifted blade again shows the streamline patterns of swept wings. In *CaseT1.0-rounded*, flow separation cannot be noticeably reduced compared to *CaseT1.0* (Fig. 9).

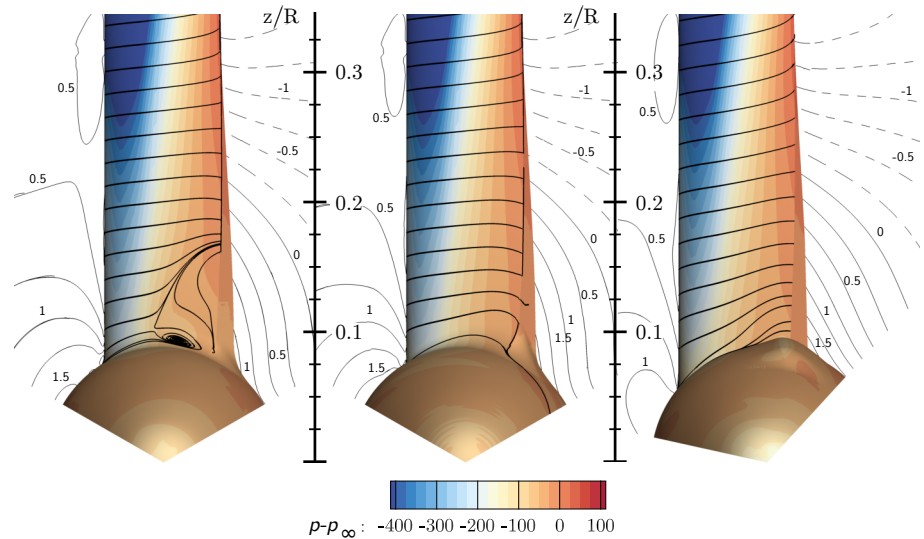

**Figure 29.** Visualization of three dimensional flow in the root region using streamlines and pressure contours for *CaseT1.0-rounded* (left), *CaseT1.0-fairing* (mid) and *CaseT1.0-dYm05* (right). The contour lines indicate the "deviation velocity" $\tilde{w}$ from to the ideal kinematic value $-\Omega y$ in a slice at $x = 2\mathrm{m}$.

More details on the separation mechanism (and suppression) can be gathered from the vortex system in the corner shown in Fig. 30. Although the footprint of the surface streamlines indicate a similar separation size for *CaseT1.0-rounded* compared to the baseline configuration with sharp juncture, recapitulation of Fig. 20 confirms that the separation thickness decreased by introducing the rounding. A side effect of the latter is a weakening of the production of the HSV in the nose region, achieved by a homogenization of the shear layer shown in the foremost slice of the approaching nacelle boundary layer interacting with the blade. Although a streamline which is counter-rotating to the HSV vortex rotating can be identified in its vicinity, no harmful corner vortex can build up and manifest itself in the junction. Separation is not initiated from a distinct streamline, but evolves from an isolated vortex generated in the transition from the rounding to the actual blade surface.

These findings were beared in mind in the design of *CaseT1.0-fairing*. It was decided to reduce the fillet nose radius to $0.2\mathrm{m}$, in order to increase the strength of the HSV again, which is believed to be helpful for reduction of corner separation (Devenport et al., 1992). This is certainly the conservative approach, since the separation should optimally be suppressed in combination with a reduced or even eliminated HSV, as the latter increases the interference drag and is a source of noise (Zess and Thole, 2001), (Simpson, 2001), (Devenport et al., 1992). Towards the trailing edge, the local radius was increased on the suction side to eliminate the unfavorable, pressure rise induced by the shape in the transition to the blade previously seen in *CaseT1.0-rounded*. As can be observed in the streamlines as well as in the foremost slice plotting the vorticity contours, the strength of the HSV could be increased by a factor of three. Its suction side leg remains closer to the blade, as it is not displaced by any recirculation and is deformed ovally when traveling downstream.

In the work of Bordji et al. (2015), the formation of the corner vortex correlated well with a peak in the Reynolds shear stress generated by the chord-wise and span-wise velocity. In order to assess the development of the corner vortex for *CaseT1.0* and *CaseT1.0-fairing*, the contours of the $\langle vw \rangle$ shear stress are depicted in Fig. 31 for a field slice cutting the blade at $x_c/c = 0.5$. The plot confirms the peak and the change in sign of shear stress for *CaseT1.0* similarly as observed in Bordji et al. (2015) and gives evidence for the production of the corner vortex. In *CaseT1.0-fairing*, the large rounding radius ensures a smooth mixing of the boundary layers resulting in a homogeneous distribution of shear stress which prevents the generation of the CV. Additionally, as it was already suggested by the streamline plot in Fig. 29, the formation of the HSV is altered, as well. Its strength is lower than in *CaseT1.0* and it is stretched. The fact that its center is located significantly closer to the blade surface is beneficial regarding its induction on the blade.

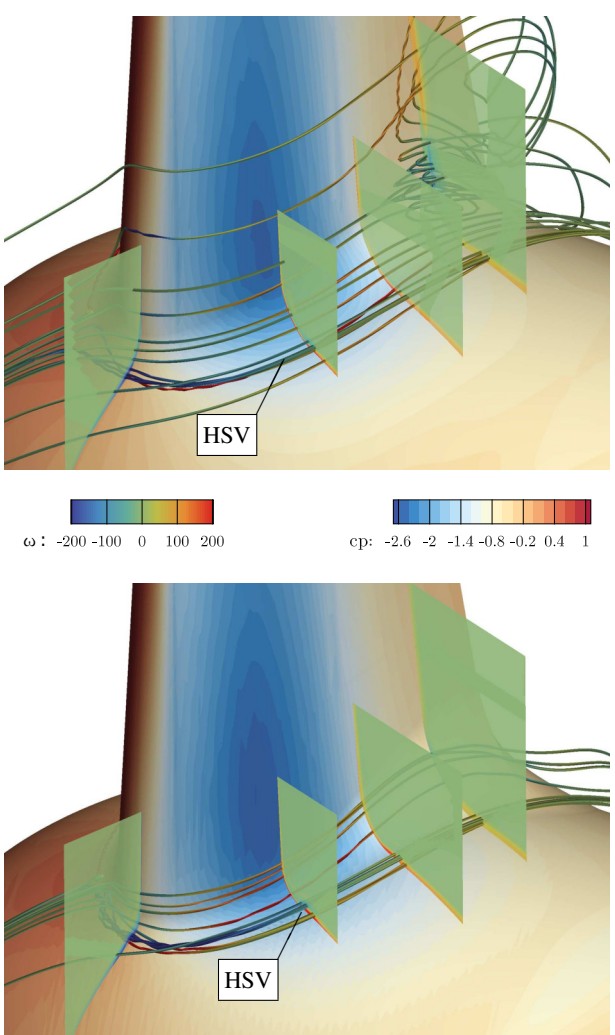

**Figure 30.** Vortex system in the junction of blade and nacelle for *CaseT1.0-rounded* (upper) and *CaseT1.0-fairing* (lower).

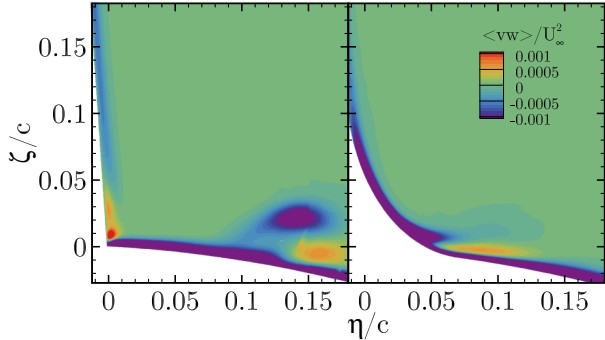

**Figure 31.** Reynolds shear stress $\langle vw \rangle$ of the chord-wise and span-wise velocity at $x_c/c = 0.5$. *CaseT1.0* (left), *CaseT1.0-fairing* (right).

**4.5.2  The Effect of Fillets on the Aerodynamic Coefficients**

Turning finally to the quantities related to aerodynamic efficiency, the pressure distribution in Fig. 32 clearly shows the improvement obtained by eliminating the corner separation for *CaseT1.0-dYm05* and *CaseT1.0-fairing*. The suction force is particularly larger between $40$-$60\%$ chord and the boundary layer is able to stand the adverse pressure gradient which leads to an effectively higher pressure value in the vicinity of the trailing edge.

The attached flow in the corner reduces the AoA (Fig. 33 c)), consistently to the previous observations, being most pronounced for *CaseT1.0-dYm05*. As already indicated by the pressure distribution in Fig. 32, *CaseT1.0-fairing* particularly increases lift and only marginally decreases drag, whereas for *CaseT1.0-dYm05* the improvements are more related to drag. Overall, tangential force increases in *CaseT1.0-fairing* due to greater lift. The evaluation of these local improvements on the global blade performance is discussed in the next section.

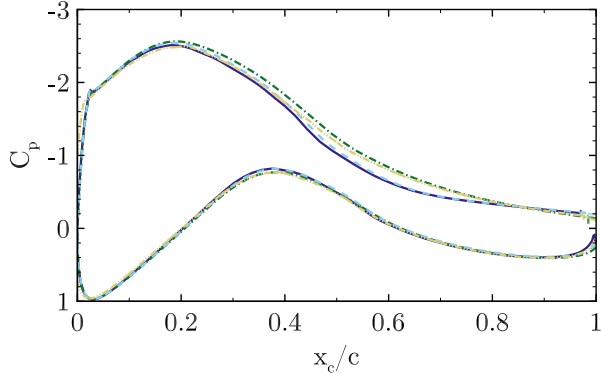

**Figure 32.** Pressure coefficient at $r/R = 0.136$. *CaseT1.0* (solid), *CaseT1.0-rounded* (dashed), *CaseT1.0-fairing* (dashed-dot), *CaseT1.0-dYm05* (dashed dot dot).

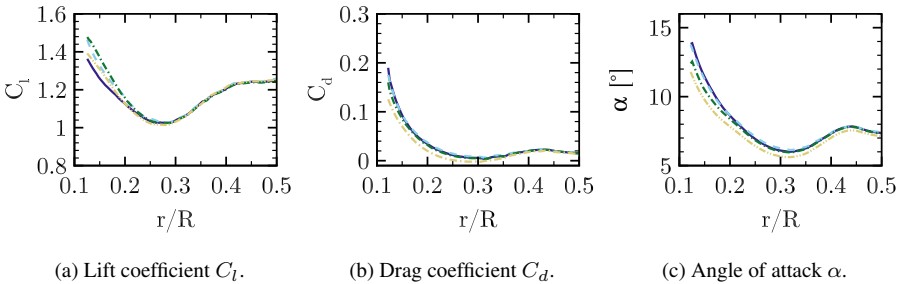

(a) Lift coefficient $C_l$.      (b) Drag coefficient $C_d$.      (c) Angle of attack $\alpha$.

**Figure 33.** Distribution of aerodynamic coefficients along the blade radius for *CaseT1.0* (solid), *CaseT1.0-rounded* (dashed) *CaseT1.0-fairing* (dashed dot) *CaseT1.0-dYm05* (dashed dot dot).

### 4.6 Assessment of Integral Quantities and Off-Design Conditions.

As previously mentioned, it is important to put the very local aerodynamic improvements in a more global context, by analyzing their impact on the sectional thrust and driving loads of the entire blade and by comparing the associated integral forces and moments. It is further important to evaluate the performance in off-design conditions. For this reason the fairing modification, which turned out to be the most promising candidate to eliminate flow separation in the junction of the blade and the nacelle, is investigated in comparison to the baseline geometry for further wind speeds, namely $U_\infty = 8; 12; 15\,\text{m/s}$. Those correspond to the tip-speed ratios of $7.48$, $5$ and $4$, respectively. Since the pitch angle is kept constant, the resulting pitch misalignments render somewhat off-design "atmospheric" conditions. The two main questions to be answered are whether the modified fairing geometry wastes performance at lower than the design wind speed and whether an additional benefit can be obtained at high wind speeds, where the overall tendency of flow separation increases. The latter fact can be important with respect to atmospheric turbulence, since gusts may cause local flow separation. If the amount of flow separation can be reduced, an overall reduction of load fluctuations could be achieved.

For the reason that unsteadiness is expected to increase with wind speed, the computations at 12 and $15\,\text{m/s}$ were continued unsteady from a steady-state solution for two more revolutions. Time-averaging of the output data was conducted for the last revolution. An overview on the effects of the wind speed on the flow field can be gained from Fig. 34, where a comparison is drawn between *CaseT1.0-fairing* and *CaseT1.0* based on the surface distributions of pressure and streamlines, as well as based on the chord-wise velocity in an airfoil section at $z/R = 0.2$. At the lowest wind speed $U_\infty = 8\,\text{m/s}$ corner separation has also almost vanished for the baseline geometry and very similar pressure, streamline and velocity distributions are obtained in both cases. For higher wind speeds the AoA increases, yielding a stronger acceleration of the flow in the front part of the airfoil and accordingly in very low surface pressures. As expected, the area of separation increases in chord-wise and radial direction. After detachment of the flow, the pressure contours flatten out in conjunction with a strong effective de-cambering of the airfoils. This is reflected in a significant recirculation area shown in the velocity contours. Overall, *CaseT1.0-fairing* significantly reduces the separated area for the high wind speed cases. Particularly, the thickness of the separated wake is

markedly reduced and the accelerated regime in the front part of the airfoil is maintained. Both result in a redirection of the airfoil wake, which consequently turns also of the aerodynamic force vector.

The resulting sectional thrust and driving loads of the blade are shown in the Figs. 35 and 36. Their integrated values resulting rotor thrust, driving force and torque (for one blade) are summarized in Tab. 1. It must be pointed out that for the lower wind speeds ($U_\infty = 8; 10\,\mathrm{m/s}$) an independent integration of each entire curve was not possible, since the effect of the fairing on the overall performance is very small. The latter fact is not surprising when keeping in mind that the flatback blade offers only little room for improvement compared to the massively separated root sections of conventional blades. The problem is

that particularly for torque, the smallest numerical uncertainties in the prediction of the forces in the outer part of the rotor can outweigh the small improvements obtained in the inner sections. Although, the results shown in the Figs. 5 and 6 imply very small influence of the grid on the solution, it must be remembered from section 3.2 that the two cases compared here cannot use exactly the same grids due topological reasons. Therefore, influences of the grid cannot be eliminated completely. Nevertheless, to be able to examine the influence of the inner portion on the overall performance, each curve was integrated

separately only up to $r/R \leq 0.35$. The result was then superposed in both cases by the same value calculated from the baseline configuration for $r/R > 0.35$. For the high wind speed cases $U_\infty = 12; 15\,\mathrm{m/s}$ each curve was independently integrated over the whole radius.

    At the lowest wind speed $U_\infty = 8\,\mathrm{m/s}$ the phenomenological impression made before is confirmed in the forces and integral quantities, which are almost identical in both cases. Hence, the first question, whether a performance degradation of the

640 fairing is present at low wind speeds can be negated. For the design wind speed $U_\infty = 10\,\mathrm{m/s}$ the local improvement of the aerodynamics discussed in the previous section is estimated to evoke a very small overall improvement of the extracted power by around $0.1\%$. When increasing the wind speed, the relative improvements compared to the baseline configuration become more pronounced. In particular, a rise of the driving force can be observed, which can be related to the relative decrease of the pressure drag caused by the separated airfoil wake. For the largest wind speed this means an overall power gain of $0.54\%$.

To summarize these findings it can be stated that it is still important to reduce the stall tendency of wind turbine blades in the inboard sections, even if it might be not a severe problem at the design point of operation, since at higher wind speeds and pitch misalignment flow separation will always expand from the root in outward direction. Therefore, the smaller the triggering flow separation near the root is, the less is the mass of separated flow that has to be transported outwards by the centrifugal pumping mechanism, which means that "clean" aerodynamic behavior can be maintained in the outer part of the rotor. With

respect to situations of very high atmospheric turbulence levels, where such temporal pitch misalignments cannot be avoided, an overall performance improvement can be expected in conjunction with a reduction of load fluctuations.

## 5   Conclusions

Numerical investigations on the flow in the root region of a wind turbine with flatback airfoils have been performed using the CFD code FLOWer.

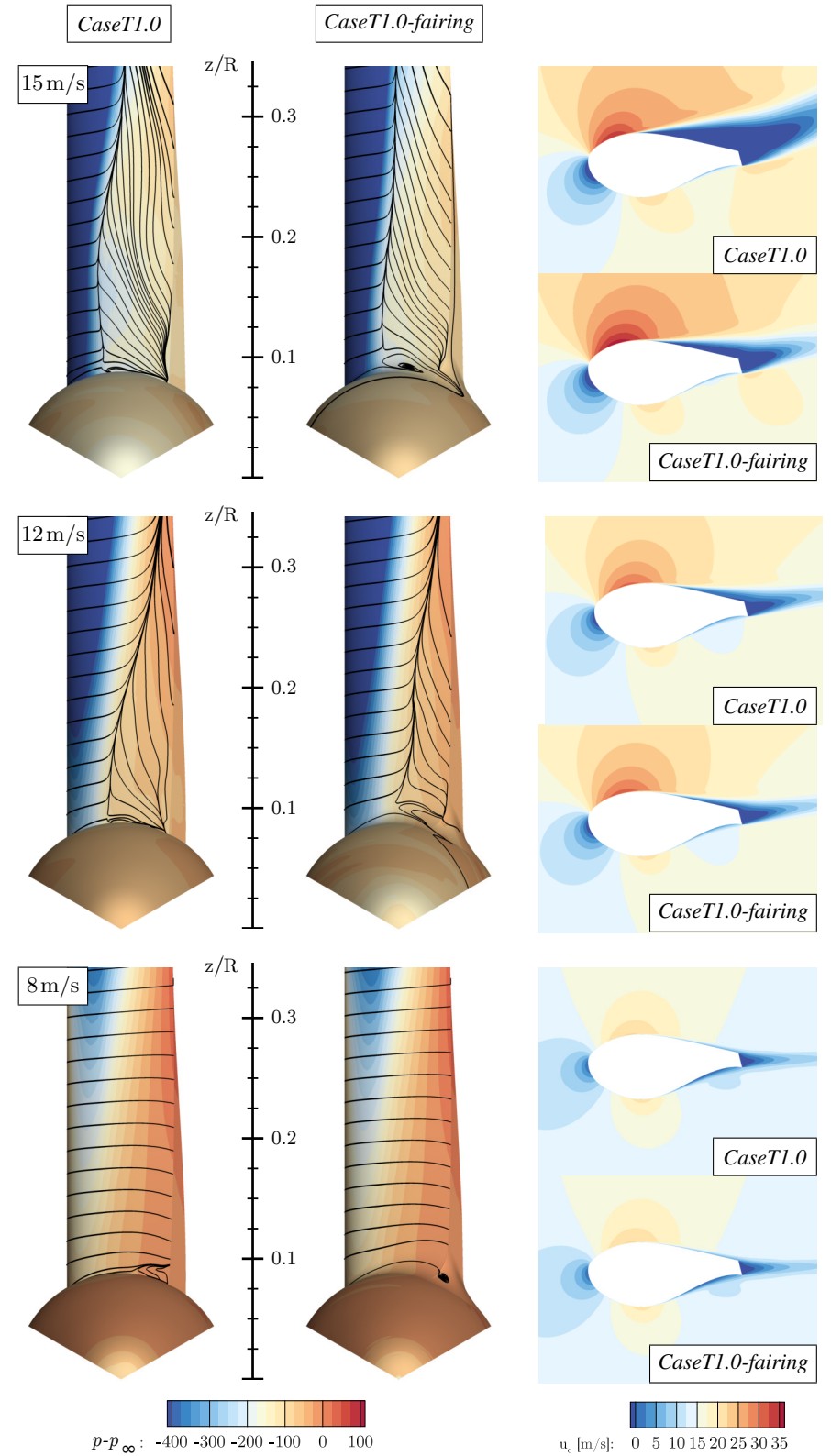

**Figure 34.** Comparison of the flow field in the root region for *CaseT1.0* and *CaseT1.0-fairing* at off-design wind speeds: $U_\infty = 8; 12; 15\,\mathrm{m/s}$. Surface streamlines (left) and chord-wise velocity distribution at $z/R = 0.2$ (right).

**Table 1.** Integral blade forces and torque for *CaseT1.0* and *CaseT1.0-fairing* at different wind speeds.

| Thrust force [kN] | | |
|---|---|---|
| Wind speed [m/s] | *CaseT1.0* | *CaseT1.0-fairing* |
| 8 | 16.442 | 16.448 (+0.040%) |
| 10 | 23.187 | 23.212 (+0.108%) |
| 12 | 28.721 | 28.850 (+0.448%) |
| 15 | 32.878 | 33.113 (+0.714%) |
| Driving force [kN] | | |
| Wind speed [m/s] | *CaseT1.0* | *CaseT1.0-fairing* |
| 8 | 1.624 | 1.628 (+0.202%) |
| 10 | 4.128 | 4.142 (+0.351%) |
| 12 | 6.936 | 7.029 (+1.336%) |
| 15 | 10.637 | 10.838 (+1.891%) |
| Torque [kNm] | | |
| Wind speed [m/s] | *CaseT1.0* | *CaseT1.0-fairing* |
| 8 | 20.309 | 20.326 (+0.082%) |
| 10 | 50.873 | 50.923 (+0.099%) |
| 12 | 84.387 | 84.833 (+0.528%) |
| 15 | 127.007 | 127.693 (+0.540%) |

**Figure 35.** Influence of different wind speeds on the sectional thrust force. *CaseT1.0* (solid), *CaseT1.0-fairing* (dashed).

The flow in the blade root region has been characterized and the influence of the nacelle on the root flow has been elaborated by comparison with an isolated rotor simulation. Flow separation in the corner of the blade and the nacelle has been identified

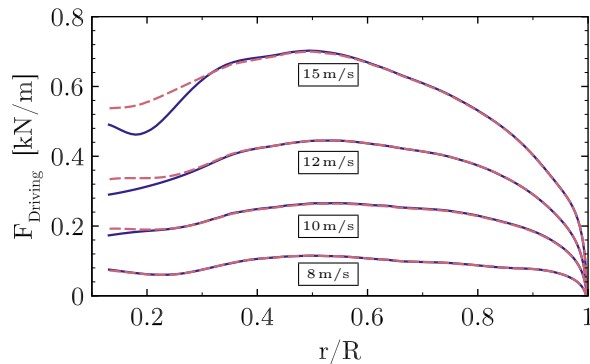

**Figure 36.** Influence of different wind speeds on the sectional driving force. *CaseT1.0* (solid), *CaseT1.0-fairing* (dashed).

as one effect that deteriorates the aerodynamic efficiency there. This flow feature is different to the commonly known massive flow separation in the root region of conventional turbines with cylindrical root sections, as it is governed by a vortex system evolving in the junction of the blade and the nacelle. Although the separation is shallow, the very high solidity of the rotor

involves strong three-dimensional effects which are determined by centrifugal and Coriolis forces as well as significant sweep angles. It can be stated that when focusing on blade root aerodynamics for those kind of rotors, a consideration of the nacelle seems to be highly important, as flow topology, aerodynamic coefficients and forces were markedly different when simulating only the isolated rotor.

Following the principal analyses, geometrical properties affecting the blade-nacelle interference have been introduced with

the objective to better understand the underlying flow physics and eventually improve the aerodynamic performance of the rotor in the root region. The first parameter investigated was the relative nacelle thickness. By increasing the nacelle thickness, flow separation in the root region drastically increased, resulting in power losses for the whole rotor of up to $1.18\%$. The reasons for the increased flow separation were found from inviscid reference simulations indicating a higher overall adverse pressure gradient for the larger nacelle thickness by means of an unfavorable interaction of the pressure minimums of the blade

and the nacelle. An important observation of the viscous simulations was also that the increased nacelle thickness intensifies the detrimental interaction of the boundary layers in the junction area. From these findings two threads for diminishing the separation have been elaborated, the first was a decoupling of the pressure gradients of the blade and the nacelle and the second aimed for an alleviation of the harmful boundary layer interaction in the junction region.

A shift of the blade forward in axial direction did not improve the stall behavior. More effective was the lateral decoupling of

the pressure gradients. By moving the blade forward in lateral direction, stall could be effectively reduced, or even eliminated when applying it to the smallest nacelle thickness, and with that, aerodynamic efficiency could be increased. The mechanisms for the improvements were explained by the decoupling of the pressure gradients of the blade and the nacelle, and by kinematic reasons which resulted in higher velocities in the front part of the blade and in the region of the adverse pressure gradient. An effective sweep angle further supported the outward transport of separated flow. The shift of the blade in the other direction

deteriorated the flow dynamics in the root region compared to the centered base version.

The attachment of fillets in the junction of the blade and the nacelle could be shown to diminish flow separation, as well. Particularly effective was the installation of a fairing, implying a small rounding radius at leading edge and a large radius near the trailing edge of the suction side. With that configuration, separation could be completely eliminated.

The comparison with the baseline geometry showed that for the design wind speed the improved aerodynamic behavior in the inner part of the rotor yields only a small improvement of the overall rotor torque. For off-design conditions it could be shown that a decrease in wind speed yields that flow separation almost disappears for the baseline geometry, as well. The fairing modification does not diminish the performance in that case. For higher wind speeds, flow separation becomes generally more severe and the relative improvements obtained by installing the fairing are more pronounced. For the latter, the area of separated wake in the very inboard region of the rotor is significantly reduced which improves whole aerodynamics for the inner one third of the blade and enhances rotor performance in the order of $0.5\%$. Hence, as an outlook, an overall performance gain and a decrease of load fluctuations can be expected for turbulent inflow conditions with high levels of turbulence intensity. Investigations on that are dedicated to future work.

Apart from such analyses, it must be stated that this study was meant to demonstrate basic parameters affecting the root flow of a wind turbine with aerodynamically shaped sections down to the hub. Potential for improvement could certainly lie in a detailed parameter optimization of any of the investigated branches. For example it could be aimed to design fillets that prevent separation and effectively alleviate the horse shoe vortex in order to decrease interference drag.

## Appendix A: The Effect of the Turbulence Model on Corner Separation

The authors are aware that the prediction of corner flow separation can be rather sensitive to the choice of the turbulence model (Vassberg et al., 2008; Rumsey et al., 2016). For this reason, a priori simulations of the simplified wing-body junction test case of Gand et al. (2015) have been performed. In principle, this is a wing mounted on a flat plate of a certain length operating at an AoA of $12°$. Three hierarchically different turbulence models have been tested; the one equation SA model (Spalart and Allmaras, 1992), the two-equation $k$-$\omega$-SST model (Menter, 1994) and the SSG-$\omega$ differential Reynolds stress model (Eisfeld, 2004). The results visualizing the corner separation are compared with experimental oil coating in Fig. A1. The SA-model expectedly to literature (Vassberg et al., 2008; Bordji et al., 2015) greatly over-predicts the amount of separation, whereas the SSG-$\omega$ model predicts hardly any separation. In turn, the SST-model provides reasonable results regarding the separation size, with a pattern quite similar to the one obtained by Bordji et al. (2015) using the SA-QCR model (Spalart, 2000). The SST-model further showed meaningful results for vortex properties (HSV and CV) and turbulent kinetic energy (Przewlocki, 2017). As the full RSM is computationally more expensive and further more delicate concerning robustness, it was decided to stick with the SST-model for the considered turbine simulations.

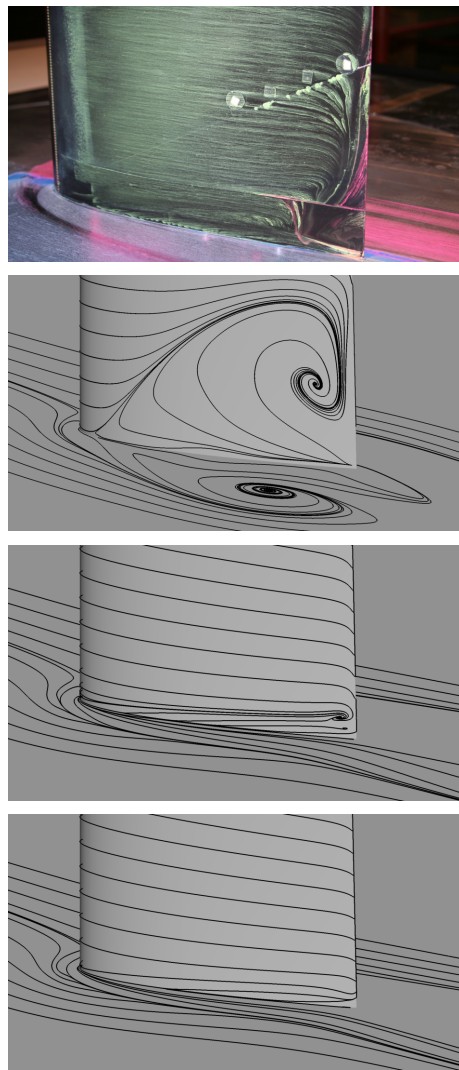

**Figure A1.** Visualization of corner separation for the simplified wing-body junction experiment of Gand et al. (2015) (upper most). Prediction with SA (upper), SST (mid), and RSM (lower). Figures reproduced from simulations of Przewlocki (2017).

## Appendix B: Assessment of the Effect of Boundary Layer Transition

The effect of boundary layer transition on the flow separation and the turbine loading is elaborated by analyzing the baseline configuration *CaseT1.0* with the transitional $\gamma$-$Re_\theta$-$k$-$\omega$-SST model, and a coupling of the SST model to the $e^N$-envelope method ($e^N$-$k$-$\omega$-SST). The critical $N$-factor is set constant to $N = 9$, and the free stream turbulence intensity entering the $\gamma$ equation of the $\gamma$-$Re_\theta$ model is adjusted such that a similar $N$-factor is obtained when applying the relation of Mack (1984). Since laminar separation bubbles are expected, the transitional computations were conducted unsteady and the sectional loads are therefore presented time-averaged.

The transition location can be identified by the strong local rise of the wall shear stress which is plotted in Fig. B1 together with the surface streamlines for the fully turbulent and transitional simulations employing the $\gamma$-$Re_\theta$ model. In the front part of the nacelle the lower shear stress suggests laminar flow. At the inner section of the rotor, transition can be identified at $x_c/c \approx 0.2$. It should be noted that for the thick airfoils considered here, the arc length of laminar flow measured from the stagnation point is considerable. Hence, the "health" of the boundary layer with laminar history is very likely better compared to the one which was assumed turbulent right from the beginning. When the boundary layer becomes subject to the adverse pressure gradient, an earlier separation can therefore be expected for fully turbulent conditions. Accordingly, it can be seen that the extent of the corner separation decreases when taking into account boundary layer transition. However, it can be also noted that the general flow topology does not change.

Turning to the sectional thrust and driving forces, cross-plotted in Fig. B2 and B3 for both, transitional and fully turbulent computations, at first glance, a good agreement of the $\gamma$-$Re_\theta$ and the $e^N$ method is obtained over the whole span. For the thrust load, the difference between transitional and fully turbulent results are most evident in the outer rotor region. In that part, the turbulent boundary layer is very thick and already indicates very small separation near the trailing edge. When the flow has a laminar history no flow separation is visible there. The boundary layer is thinner which reduces the effective de-cambering of the airfoils so that they can generate more lift and have less drag. The latter increases the driving force. In the inner section of the rotor, the same mechanism holds which means increasing lift by alleviation of the stress to the boundary layer. Since the twist angles are large, the increasing lift is distinctively reflected in the driving force, as well.

It can be concluded that expectedly the accounting for boundary layer transition increases thrust and driving forces of the rotor, by increasing lift and reducing viscous drag. The span-wise and chord-wise extent of the corner separation is reduced, but no fundamental change of the topology is visible. As suggested by measurements such as those of Zamir (1981), transition in the corner region is typically earlier than for an equivalent flat plate boundary layer. Hence in reality, where also pollution and surface roughness plays a role, the extent of separation might lie in between the one predicted by the fully turbulent and the transitional simulation. Both facts justify the fully turbulent approach when comparing the different geometry modifications.

## Appendix C:  Nomenclature

*Competing interests.*  The authors declare that they have no conflict of interest.

*Acknowledgements.* The authors acknowledge the *German Federal Ministry for Economic Affairs and Energy* for funding this study as part of the project "AssiSt" (grant number 0325719A) and the *High Perfomance Computing Center Stuttgart* for the contribution of the computational resources.

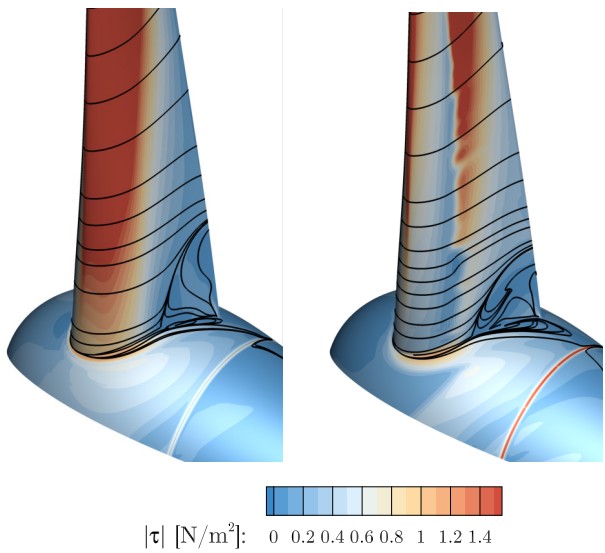

**Figure B1.** Surface streamlines and contours of shear stress magnitude for fully turbulent $k$-$\omega$-SST simulation (left) and $\gamma$-$Re_\theta$-$k$-$\omega$-SST simulation (right).

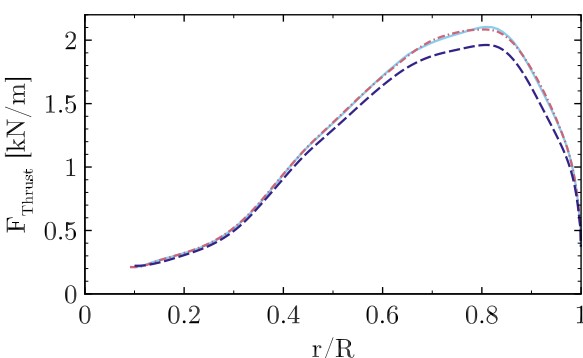

**Figure B2.** Influence of boundary layer transition on the sectional thrust force. Fully turbulent $k$-$\omega$-SST (dashed), $\gamma$-$Re_\theta$-$k$-$\omega$-SST (solid), $e^N$-$k$-$\omega$-SST (dashed dot).

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

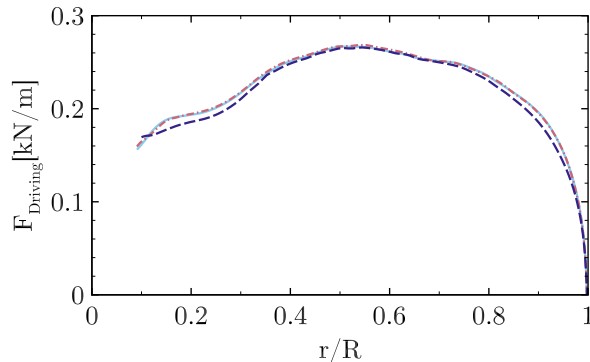

**Figure B3.** Influence of boundary layer transition on the sectional driving force. Fully turbulent $k$-$\omega$-SST (dashed), $\gamma$-$Re_\theta$-$k$-$\omega$-SST (solid), $e^N$-$k$-$\omega$-SST (dashed dot).

Baldacchino, D., Ferreira, C., Tavernier, D. D., Timmer, W., and van Bussel, G.: Experimental parameter study for passive vortex generators on a 30% thick airfoil, Wind Energy, 2018.

Bangga, G.: Three-Dimensional Flow in the Root Region of Wind Turbine Rotors, kassel university press GmbH, 2018.

Bangga, G., Lutz, T., Jost, E., and Krämer, E.: Erratum:"CFD studies on rotational augmentation at the inboard sections of a 10 MW wind turbine rotor"[J. Renewable Sustainable Energy 9, 023304 (2017)], Journal of Renewable and Sustainable Energy, 10, 019 902, 2018.

Benek, J. A., Steger, J. L., Dougherty, F. C., and Buning, P. G.: in: Chimera. A Grid-Embedding Technique., 1986.

Boorsma, K., Schepers, J., Gomez-Iradi, S., Herraez, I., Lutz, T., Weihing, P., Oggiano, L., Pirrung, G., Madsen, H., Shen, W., Rahimi, H., and Schaffarczyk, P.: Final report of IEA Task 29, Mexnet (Phase 3), IEA, 2018.

Bordji, M., Gand, F., Deck, S., and Brunet, V.: Investigation of a Nonlinear Reynolds-Averaged Navier–Stokes Closure for Corner Flows, AIAA Journal, 2015.

Devenport, W. J., Simpson, R. L., Dewitz, M. B., and Agarwal, N. K.: Effects of a leading-edge fillet on the flow past an appendage-body junction, AIAA journal, 30, 2177–2183, 1992.

Drela, M. and Giles, M. B.: Viscous-inviscid analysis of transonic and low Reynolds number airfoils, AIAA journal, 25, 1347–1355, 1987.

Du, Z. and Selig, M.: The effect of rotation on the boundary layer of a wind turbine blade, Renewable Energy, 20, 167–181, 2000.

Dwyer, H. and Aiccroskey, W.: Crossflow and unsteady boundary-layer effects on rotating blades, AIAA Journal, 9, 1498–1505, 1971.

Eisfeld, B.: Implementation of Reynolds stress models into the DLR-FLOWer code, 2004.

Gand, F., Monnier, J.-C., Deluc, J.-M., and Choffat, A.: Experimental study of the corner flow separation on a simplified junction, AIAA Journal, 2015.

Grabe, C. and Krumbein, A.: Extension of the $\gamma$-Re$\theta$t model for prediction of crossflow transition, in: 52nd Aerospace Sciences Meeting, p. 1269, 2014.

Guntur, S. and Sørensen, N. N.: A study on rotational augmentation using CFD analysis of flow in the inboard region of the MEXICO rotor blades, Wind Energy, 18, 745–756, 2015.

Herráez, I., Stoevesandt, B., and Peinke, J.: Insight into rotational effects on a wind turbine blade using Navier–Stokes computations, Energies, 7, 6798–6822, 2014.

Herráez, I., Akay, B., van Bussel, G. J., Peinke, J., and Stoevesandt, B.: Detailed analysis of the blade root flow of a horizontal axis wind turbine, Wind Energy Science, 1, 89–100, 2016.

Himmelskamp, H.: Profile investigations on a rotating airscrew, MAP, 1947.

Jameson, A.: Time dependent calculations using multigrid, with applications to unsteady flows past airfoils and wings, AIAA paper, 1596, 1991, 1991.

Jameson, A., Schmidt, W., Turkel, E., et al.: Numerical solutions of the Euler equations by finite volume methods using Runge-Kutta time-stepping schemes, AIAA paper, 1259, 1981, 1981.

Johansen, J. and Sørensen, N. N.: Aerofoil characteristics from 3D CFD rotor computations, Wind Energy, 7, 283–294, 2004.

Johansen, J., Madsen, H. A., Sørensen, N., and Bak, C.: Numerical Investigation of a Wind Turbine Rotor with an aerodynamically redesigned hub-region, in: 2006 European wind energy conference and exhibition, Athens, Greece, 2006.

Jost, E., Klein, L., Leiprand, H., Lutz, T., and Krämer, E.: Extracting the angle of attack on rotor blades from CFD simulations, Wind Energy. Accepted, 2018.

Klein, L., Gude, J., Wenz, F., Lutz, T., and Krämer, E.: Advanced CFD-MBS coupling to assess low-frequency emissions from wind turbines, Wind Energy Science Discussions, 2018, 1–30, https://doi.org/10.5194/wes-2018-51, https://www.wind-energ-sci-discuss.net/wes-2018-51/, 2018.

Knezevici, D., Sjolander, S., Praisner, T., Allen-Bradley, E., and Grover, E.: Measurements of secondary losses in a turbine cascade with the implementation of nonaxisymmetric endwall contouring, Journal of Turbomachinery, 132, 011 013, 2010.

Kowarsch, U., Keßler, M., and Krämer, E.: High order CFD-simulation of the rotor-fuselage interaction, 2013.

Kroll, N., Rossow, C.-C., Becker, K., and Thiele, F.: The MEGAFLOW project, Aerospace Science and Technology, 4, 223–237, 2000.

Kühn, T., Altmikus, Daboul, H., Radi, A., Raasch, S., Knigge, C., Böske, L., Schwarz, T., Heister, C., Möller, A., Lutz, T., Weihing, P.,
Schulz, C., Thiemeier, J., Mockett, C., Fuchs, M., and Thiele, F.: AssiSt-Schlussbericht gemäß NKBF98, Tech. rep., 2018.

Langtry, R.: Extending the Gamma-Rethetat Correlation Based Transition Model for Crossflow Effects, in: 45th AIAA Fluid Dynamics Conference, p. 2474, 2015.

Langtry, R. B. and Menter, F. R.: Correlation-based transition modeling for unstructured parallelized computational fluid dynamics codes, AIAA journal, 47, 2894–2906, 2009.

Leishman, G. J.: Principles of helicopter aerodynamics with CD extra, Cambridge university press, 2006.

Leishman, J.: Modeling sweep effects on dynamic stall, Journal of the American Helicopter Society, 34, 18–29, 1989.

Letzgus, J., Dürrwächter, L., Schäferlein, U., Keßler, M., and Krämer, E.: Optimization and HPC-Applications of the Flow Solver FLOWer, in: High Performance Computing in Science and Engineering'17, pp. 305–322, Springer, 2018.

Levy, D. W., Laflin, K. R., Tinoco, E. N., Vassberg, J. C., Mani, M., Rider, B., Rumsey, C. L., Wahls, R. A., Morrison, J. H., Brodersen, O. P.,
et al.: Summary of data from the fifth computational fluid dynamics drag prediction workshop, Journal of Aircraft, 2014.

Lindenburg, C.: Investigation into rotor blade aerodynamics, ECN Report: ECN-C-03-025, 2003.

Loganathan, J. and Gopinath, G.: Advances in Wind Turbine Aerodynamics, 2018.

Mack, L. M.: Boundary-layer linear stability theory, Tech. rep., CALIFORNIA INST OF TECHNOLOGY PASADENA JET PROPULSION LAB, 1984.

Masson, C. and Smaïli, A.: Numerical study of turbulent flow around a wind turbine nacelle, Wind Energy, 9, 281–298, 2006.

McCormick, D.: Boundary layer separation control with directed synthetic jets, in: 38th Aerospace Sciences Meeting and Exhibit, p. 519, 2000.

Menter, F. R.: Two-equation eddy-viscosity turbulence models for engineering applications, AIAA journal, 32, 1598–1605, 1994.

Obert, E.: Aerodynamic design of transport aircraft, IOS press, 2009.

Post, M. L. and Corke, T. C.: Separation control on high angle of attack airfoil using plasma actuators, AIAA journal, 42, 2177–2184, 2004.

Przewlocki, J.: Numerische Simulation einer vereinfachten Flügel-Rumpf Konfiguration, Master's thesis, University of Stuttgart, Institute of
      Aerodynamics and Gas Dynamics, 2017.

Purser, P. E. and Spearman, M. L.: Wind-tunnel tests at low speed of swept and yawed wings having various plan forms, Tech. rep., National
      Aeronautics And Space Administration Hampton VA Langley Research Center, 1951.

Rahimi, H., Schepers, G., Shen, W. Z., García, N. R., Schneider, M., Micallef, D., Ferreira, C. S., Jost, E., Klein, L., and Herráez, I.:
      Evaluation of different methods for determining the angle of attack on wind turbine blades with CFD results under axial inflow conditions,
      arXiv preprint arXiv:1709.04298, 2017.

Rumsey, C. L., Neuhart, D., and Kegerise, M. A.: The NASA juncture flow experiment: Goals, progress, and preliminary testing, in: 54th
      AIAA Aerospace Sciences Meeting, p. 1557, 2016.

Sayed, M., Lutz, T., Krämer, E., Shayegan, S., Ghantasala, A., Wüchner, R., and Bletzinger, K.-U.: High fidelity CFD-CSD aeroelastic
      analysis of slender bladed horizontal-axis wind turbine, in: Journal of Physics: Conference Series, vol. 753, p. 042009, IOP Publishing,
      2016.

Schepers, J. and Snel, H.: Model experiments in controlled conditions, ECN report, 2007.

Schepers, J., Boorsma, K., Cho, T., Gomez-Iradi, S., Schaffarczyk, P., Jeromin, A., Lutz, T., Meister, K., Stoevesandt, B., Schreck, S., et al.:
Final report of IEA Task 29, Mexnet (Phase 1): analysis of Mexico wind tunnel measurements, IEA, 2012.

Schreck, S. and Robinson, M.: Rotational augmentation of horizontal axis wind turbine blade aerodynamic response, Wind Energy, 5, 133–
      150, 2002.

Schreck, S., Fingersh, L., Siegel, K., Singh, M., and Medina, P.: Rotational augmentation on a 2.3-MW rotor blade with thick flatback airfoil
      cross sections, in: Proceedings of the 51st AIAA Aerospace Sciences Meeting, AIAA 2013, vol. 915, 2013.

Seifert, A., Bachar, T., Koss, D., Shepshelovich, M., and Wygnanskil, I.: Oscillatory blowing: a tool to delay boundary-layer separation,
      AIAA journal, 31, 2052–2060, 1993.

Simpson, R. L.: Junction flows, Annual Review of Fluid Mechanics, 33, 415–443, 2001.

Snel, H., Houwink, R., Bosschers, J., Piers, W., Van Bussel, G., and Bruining, A.: Sectional prediction of 3D effects for stalled flow on rotating
      blades and comparison with measurements, in: Proc. European Community Wind Energy Conference, HS Stevens and Associates, LÃ1,
840   vol. 4, 1993.

Sørensen, N., Hansen, M., Garca, N., Florentie, L., Boorsma, K., Gomez-Iradi, S., Prospathopoulos, J., Papadakis, G., Voutsinas, S., Barakos,
      G., et al.: Power curve predictions wp2 deliverable 2.3, Technical Report, 2014.

Sørensen, N., Garca, N., Voutsinas, S., Jost, E., and Lutz, T.: Aerodynamics of Large Rotors WP2 Deliverable 2.6 Effects of complex inflow
      for the AVATAR reference rotor and NM80 rotors, Technical Report, 2017.

Spalart, P. and Allmaras, S.: A one-equation turbulence model for aerodynamic flows, in: 30th aerospace sciences meeting and exhibit, p.
      439, 1992.

Spalart, P. R.: Strategies for turbulence modelling and simulations, International Journal of Heat and Fluid Flow, 21, 252–263, 2000.

Tanner, M.: Reduction of base drag, Progress in Aerospace Sciences, 16, 369–384, 1975.

Thwaites, B.: Approximate calculation of the laminar boundary layer, The Aeronautical Quarterly, 1, 245–280, 1949.

Van Dam, C., Kahn, D. L., and Berg, D. E.: Trailing edge modifications for flatback airfoils, SAND2008-1781, Sandia National Laboratories, Albuquerque, NM, 2008.

Vassberg, J., Tinoco, E., Mani, M., Rider, B., Zickuhr, T., Levy, D., Brodersen, O., Eisfeld, B., Crippa, S., Wahls, R., et al.: Summary of the Fourth AIAA CFD Drag Prediction Workshop (2010). AIAA Paper No, Tech. rep., AIAA-2010-4547, 2010.

Vassberg, J. C., Tinoco, E. N., Mani, M., Brodersen, O. P., Eisfeld, B., Wahls, R. A., Morrison, J. H., Zickuhr, T., Laflin, K. R., and Mavriplis,
D. J.: Abridged summary of the third AIAA computational fluid dynamics drag prediction workshop, Journal of Aircraft, 45, 781–798, 2008.

Weihing, P., Letzgus, J., Bangga, G., Lutz, T., and Krämer, E.: Hybrid RANS/LES capabilities of the flow solver FLOWer-application to flow around wind turbines, in: The 6th Symposium on Hybrid RANS-LES Methods, Strassbourg, 2016.

Weihing, P., Schulz, C., Lutz, T., and Krämer, E.: Comparison of the Actuator Line Model with Fully Resolved Simulations in Complex
Environmental Conditions, in: Journal of Physics: Conference Series, vol. 854, p. 012049, IOP Publishing, 2017.

Zahle, F. and Sørensen, N. N.: Characterization of the unsteady flow in the nacelle region of a modern wind turbine, Wind Energy, 14, 271–283, 2011.

Zamir, M.: Similarity and stability of the laminar boundary layer in a streamwise corner, Proc. R. Soc. Lond. A, 377, 269–288, 1981.

Zess, G. and Thole, K.: Computational design and experimental evaluation of using a leading edge fillet on a gas turbine vane, in: ASME
Turbo Expo 2001: Power for Land, Sea, and Air, pp. V003T01A083–V003T01A083, American Society of Mechanical Engineers, 2001.

**Table C1.** Nomenclature

| Symbol | Unit | Description |
|---|---|---|
| $c$ | $[m]$ | Airfoil chord length |
| $C_d$ | $[-]$ | Drag coefficient |
| $C_{f,x}$ | $[-]$ | Chord-wise skin friction coefficient |
| $C_l$ | $[-]$ | Lift coefficient |
| $C_p$ | $[-]$ | Pressure coefficient |
| $d_w$ | $[m]$ | Wall distance |
| $L$ | $[m]$ | Length of the nacelle |
| $N$ | $[-]$ | Critical amplification factor of Tollmien-Schlichting waves |
| $p$ | $[Pa]$ | Local static pressure |
| $p_\infty$ | $[Pa]$ | Ambient pressure |
| $q_{\infty,rot}$ | $[Pa]$ | Kinematic stagnation pressure in the rotating frame |
| $r$ | $[m]$ | Local radius |
| $R$ | $[m]$ | Radius of the blade |
| $u$ | $[m/s]$ | Axial velocity |
| $u_c$ | $[m/s]$ | Velocity in chord-wise direction |
| $u_{mag}$ | $[m/s]$ | Velocity magnitude |
| $U_\infty$ | $[m/s]$ | Wind speed |
| $U_{\infty,rot}$ | $[m/s]$ | Kinematic inflow velocity in the rotating frame |
| $v$ | $[m/s]$ | Lateral velocity in the rotating frame |
| $w$ | $[m/s]$ | Vertical velocity in the rotating frame |
| $\langle vw \rangle$ | $[m^2/s^2]$ | Reynolds shear stress |
| $\tilde{w}$ | $[m/s]$ | Vertical motion relative to the ideal circular path. |
| $x$ | $[m]$ | Axial coordinate |
| $y$ | $[m]$ | Lateral coordinate in the rotating frame |
| $y_p^+$ | $[-]$ | Non-dimensional wall distance wall adjacent cell |
| $z$ | $[m]$ | Vertical coordinate in the rotating frame |
| $x_c$ | $[m]$ | Coordinate in chord-wise direction |
| $\alpha$ | $[-]$ | Angle of attack |
| $\Gamma$ | $[m^2/s]$ | Bound circulation |
| $\eta$ | $[m]$ | Blade-normal coordinate in the corner frame |
| $\rho$ | $[kg/m^3]$ | Density |
| $\Omega$ | $[1/s]$ | Rotational speed |
| $\omega$ | $[1/s]$ | Vorticity |
| $\zeta$ | $[m]$ | Vertical coordinate in the corner frame |