# Peer review of "Numerical Analyses and Optimizations on the Flow in the Nacelle Region of a Wind Turbine"

_Wind Energy Science, 2018_

## Referee Comment (RC1) · A. Smaili (Referee) · 8 Jun 2018

A. Smaili (Referee)

arezki.smaili@g.enp.edu.dz

This paper deals with numerical study of the flow field features within the hub region of wind turbine rotor. The study is focusing on the effect of nacelle geometry upon blade root aerodynamics. For this purpose, the generic version of the Enercon E44 wind turbine nacelle integrating blades with flatback airfoils root has been considered. The finite volume solver of FLOWer code based on compressible Navier-Stokes equations has been used. The manuscript is well presented and organised. However, I have the following remarks.

1. The flow in the nacelle region of a wind turbine has been the subject of several previous studies. Thereby, the authors may improve the literature review, a more serious

bibliographical search and study might be carried out. 2. The mesh study was not presented: the choice of grid type and size was not justified. 3. The use of a compressible Navier-Stokes solver should be justified. It would be desirable to present the contours of the Mach number. 4. The validity of the numerical simulations was not presented. 5. The full-turbulence models are not suitable for describing the flow fields in the hub and nacelle region; because probably in such situation, boundary layer transition may occur, and therefore conclusions drawn might be far from reality. 6. To obtain more relevant conclusions, the simulations should be carried out for other wind speed values, not only for 10 m/s.

---

## Referee Comment (RC2) · Anonymous Referee #2 · 11 Jun 2018

The manuscript is a numerical study of the flow dynamic of a wind turbine rotor, focusing on the root region. A very detailed view of the complex flow induced in this region is presented. Aerodynamic losses related to the vortex system and to flow separation could be important. The authors presented 3 strategies related to the nacelle in order to increase the aerodynamic efficiency in the root region. The paper is generally clear, well written and well structured. My comments: 1. In the introduction, it would be useful to mention other techniques that can be used to improve aerodynamic efficiency in the root region with flatback airfoil, such as flow control devices (splitter, cavity, flap...), and to show how the authors' solutions can stand out. 2. With no validation of the model and without a grid independency study, how can you be sure about the results accuracy? 3. Are there any manufacturing constraints for the proposed solutions to

increase aerodynamic efficiency? 4. The proposed nacelle modifications and their impacts locally, on the root flow have been shown in detail. However, it would be interesting to quantify the impact of these solutions on a global parameter performance, such as the total power produced of the turbine.

---

## Author Comment (AC1) · 17 Jul 2018

Dear Prof. Smaili, please find attached two documents, which contain the responses on your comments, as well as a marked-up version of the revised manuscript. Kind regards, Pascal Weihing

Please also note the supplement to this comment:
https://www.wind-energ-sci-discuss.net/wes-2018-38/wes-2018-38-AC1-supplement.zip

---

## Author Response (AR1)

**This document includes:**

1. Point-to-point response to the first reviewer

2. Point-to-point response to the second reviewer

3. List of the major changes in the manuscript

4. Marked-up manuscript
   - changed sections with regard to the comments by reviewer 1 are marked in yellow
   - changed sections with regard to the comments by reviewer 2 are marked in cyan
   - changed sections with regard to the comments by both reviewers are marked in green
   - changes with regard to no comments but which serve a better understanding and an improvement of the manuscript are marked in gray

**Reply to comments by Reviewer 1**

Pascal Weihing on behalf of the authors
IAG, University of Stuttgart

July 17, 2018

The authors would like to thank Prof. Smaili for his efforts and valuable comments. They are very much appreciated and incorporated into the revised paper.

In the present document the comments given by the 1st reviewer are addressed consecutively. The following formatting is chosen:

- The reviewer comments are marked in blue and italic.

- The reply by the authors is in black color

- A marked-up manuscript is added. Changed section with regard to the comments by reviewer 1 are marked in yellow. Changed sections with regard to comments by both reviewers are marked in green. Highlighting in gray denotes passages that have been changed by the authors in order to improve the clarity or the argumentation but which are not related to specific reviewer comments.

**General comments**

1. "*The flow in the nacelle region of a wind turbine has been the subject of several previous studies. Thereby, the authors may improve the literature review, a more serious bibliographical search and study might be carried out.*"

A new subsection has been added in the introduction which addresses the interacting flow fields of the rotor and the nacelle $\boxed{\textbf{R1:G1}}$ (page 3, line 66). The following references have been added:

Masson, C., & Smaïli, A. (2006). Numerical study of turbulent flow around a wind turbine nacelle. Wind Energy: An International Journal for Progress and Applications in Wind Power Conversion Technology, 9(3), 281-298.

Zahle, F., & Sørensen, N. N. (2011). Characterization of the unsteady flow in the nacelle region of a modern wind turbine. Wind Energy, 14(2), 271-283.

Johansen, J., Madsen, H. A., Sørensen, N. N., & Bak, C. (2006). Numerical Investigation of a Wind Turbine Rotor with an aerodynamically redesigned hub-region. In 2006 European wind energy conference and exhibition, Athens, Greece.

2. "*The mesh study was not presented: the choice of grid type and size was not justified*"

The authors agree that an assessment of the accuracy of numerical predictions is very important, particularly if there is no reference data available to validate the results. The grids in the present study are based on experiences gained during many national (AssiST, DFG-PAK780, LARS, TremAc, OWEALoads) and international research projects (MexNext, AVATAR, Innwind) and are based on the recommendations for the cell spacings and growth rates made during the NASA drag prediction work shops. In order to check for the influence of the grid on the solution in the present study a very fine grid has been employed for comparison which is planned to be used for future DES simulation. The trend on the sectional load distribution shows that there is only a very small grid influence. The sectional thrust curves more or less collapse completely. For the sectional driving force, very small deviations are visible in the radial distribution. Interestingly, the differences in the integral driving force is one order of magnitude smaller compared to the differences in the integral thrust. For this reason a classical grid convergence study (GCI) can be sometimes misleading, since local effects might be caught up by error compensation. However, these local effects are particularly important the present case where a detailed analyses of three-dimensional features are studied. Probably, a "bad" grid in the root region with for example large skew angles, aspect ratios or under-resolved boundary layers would not allow allow for a detailed evaluation of the relevant flow features, but on the other hand would also not be reflected in a global GCI.

Although, the general impact of the grid on the solution seems to be very small, it must be noted that for the lower wind speeds the local effects of the aerodynamic modifications on the overall blade performance can come into the same order of magnitude as the accuracy of the CFD framework. This fact is analyzed in the newly introduced section 4.6.

Regarding the grid dependency analysis, the paper has been modified in $\boxed{\text{R1:G2-a}}$ (page 7, line 181) and $\boxed{\text{R1:G2-b}}$ (page 10, line 224).

3. "*The use of a compressible Navier-Stokes solver should be justified. It would be desirable to present the contours of the Mach number.* "

It is certainly clear that the use of a compressible flow solver is not necessary when dealing with flow features in the very inboard region of a wind turbine. In the present cases the maximum Mach numbers on the suction side of the airfoils in the tip region was around 0.29. The simple reason, why we use this solver is that it is the only available one within the code FLOWer. FLOWer on the other hand is a well proven CFD code that has been applied in numerous aerodynamic and aeroelastic studies of wind and helicopter applications, so that a lot of experience has been gained. The code is continuously further developed at the authors institute. A list of references can be found at
`https://www.iag.uni-stuttgart.de/abteilungen/luftfahrzeugaerodynamik/veroeffentl_luftf/veroeffentlichungen_luftf.index.html`

An additional fact to add is that in the future, compressibility effects will play a more important role, when the turbine diameters increase or when for example offshore higher tip-speeds might be realized. For the high tip speed cases of for example the DTU 10MW rotor, or even for the small scale MEXICO turbine over-predictions of the lift in the outer portion of the rotor at higher angles of attack predicted by some incompressible codes could be explained by the neglection of compressibility. When taking into account compressibility the adverse pressure gradient increases which leads to an earlier separation compared to an incompressible assumption. An investigation on that has been a task of the AVATAR project, where a comparison of FLOWer and the EllipSys code focusing on the effects of compressibility and possible corrections has been conducted and which was recently presented within a collaborative paper.

Sørensen, N. N., Bertagnolio, F., Jost, E., & Lutz, T. (2018, June). Aerodynamic effects of compressibility for wind turbines at high tip speeds. In Journal of Physics: Conference Series (Vol. 1037, No. 2, p. 022003). IOP Publishing.

4. "*The validity of the numerical simulations was not presented.*"

This is correct, since no measurement data was available for validation. Additional references have been placed in section 3.1 $\boxed{\text{R1:G4}}$ (page 5, line 144), which state that the present numerical methodology has given accurate results in other projects, where measurements were available and code-to-code comparisons have been performed.

5. "*The full-turbulence models are not suitable for describing the flow fields in the hub and nacelle region; because probably in such situation, boundary layer transition may occur, and therefore conclusions drawn might be far from reality.* "

The authors agree that boundary layer transition might affect the development of flow separation in the hub region. Accounting for the laminar flow history in the front part of the blade will probably lead to a downstream shift of the separation, as the shear stress of the "freshly" transitioned boundary layer is higher compared to the boundary layer which was turbulent right from the beginning. In order evaluate this hypothesis and to check whether the entire flow pattern changes, or not, transitional simulations have been performed for the baseline geometry. Two transition models have been chosen to draw a comparison with the fully turbulent results: The correlation based $\gamma$-$Re_\theta$ model as well and $e^N$ envelope model, both coupled to SST closure. The results are presented in Appendix B. As expected, flow separation diminishes by including the effect of transition. However, the main flow topology stays the same. For the production runs comparing the different geometries, boundary layer transition was omitted or the following reasons:

- Reduction of additional model uncertainty: The effects induced by the geometrical modifications on the overall rotor performance is rather small. Additional uncertainties stemming for example from slight deviations in the transition location (which might be unsteady) are unfavorable.

- There is no engineering transition model that can accurately account for the mechanisms describing boundary layer transition in corner flows. In that region strong cross flow prevails which neither the state-of-the art $\gamma$-$Re_\theta$ model nor the $e^N$ envelope method accounts for.

- Experimental studies such as those of

  Zamir, M. "Similarity and stability of the laminar boundary layer in a streamwise corner." Proc. R. Soc. Lond. A 377.1770 (1981): 269-288

  suggest that boundary layer transition in corner flows occurs earlier than in equivalent conditions over a flat plate.

- In reality, pollution and erosion leads to an earlier transition than predicted by the standard models that do not include roughness effects.

Hence, with respect to the latter two points the actual transition location and the flow field development can be expected to be somewhat in between the fully turbulent simulations and the transitional cases presented in Appendix B. These deliberations can be found in the revised paper in $\boxed{\text{R1:G5-a}}$ (page 5, line 154), whereas the results comparing fully turbulent and transition simulations are presented in $\boxed{\text{R1:G5-b}}$ (page 38, line 711)

6. "*To obtain more relevant conclusions, the simulations should be carried out for other wind speed values, not only for 10 m/s.*"

Thank you for pointing that out. The additional wind speeds 8, 12 and 15m/s have been analyzed for the baseline and the optimized geometry. In particular for the higher wind speeds valuable information on the stall mechanisms and on the global load behavior could be deduced. These additional cases can be found in the newly introduced section 4.6. The relevant text passages added are $\boxed{\textbf{R1:G6-a}}$ (page 9, line 216), $\boxed{\textbf{R1:G6-b}}$ (page 33, line 602) and $\boxed{\textbf{R1:G6-c}}$ (page 38, line 684).

**Reply to comments by Reviewer 2**

Pascal Weihing on behalf of the authors
IAG, University of Stuttgart

July 17, 2018

The authors would like to thank the reviewer for his/her efforts and valuable comments. They are very much appreciated and incorporated into the revised paper.

In the present document the comments given by the 2nd reviewer are addressed consecutively. The following formatting is chosen:

- The reviewer comments are marked in blue and italic.

- The reply by the authors is in black color

- A marked-up manuscript is added. Changed section with regard to the comments by reviewer 2 are marked in cyan. Changed sections with regard to comments by both reviewers are marked in green. Highlighting in gray denotes passages that have been changed by the authors in order to improve the clarity or the argumentation but which are not related to specific reviewer comments.

**General comments**

1. "*In the introduction, it would be useful to mention other techniques that can be used to improve aerodynamic efficiency in the root region with flatback airfoil, such as flow control devices (splitter, cavity, flap...), and to show how the authors' solutions can stand out.*"

This point has been added to the introduction in section $\boxed{\textbf{R2:G1-a}}$ (page 3, line 81) by mentioning measures to increase the efficiency of conventional root sections and root sections with flatback airfoils. For the cylinder like sections active and passive flow control technologies such as VGs, or blowing/plasma actuators might be utilized in future applications. For flatback airfoil sections the causes of the base drag are briefly described and possible solutions are listed such as Gourney flaps, splitter plates, or cavities. The relevant literature describing all these measures is given. In addition, a clearer distinction of the scope of the present work from these efficiency boosting technologies is given in section 1.5. $\boxed{\textbf{R2:G1-b}}$ (page 4, line 97)

2. "*With no validation of the model and without a grid independency study, how can you be sure about the results accuracy?*"

This point was also argued by Reviewer#1, so that a critical examination of this has been conducted. The authors agree that an assessment of the accuracy of numerical predictions is very important, particularly if there is no reference data available to validate the results. The grids in the present study are based on experiences gained during many national (AssiST, DFG-PAK780, LARS, TremAc, OWEALoads) and international research projects (MexNext, AVATAR, Innwind) and are based on the recommendations for the cell spacings and growth rates made during the NASA drag prediction work shops. In order to check for the influence of the grid on the solution in the present study a very fine grid has been employed for comparison which is planned to be used for future DES simulation. The trend on the sectional load distribution shows that there is only a very small grid influence. The sectional thrust curves more or less collapse completely. For the sectional driving force, very small deviations are visible in the radial distribution. Interestingly, the differences in the integral driving force is one order of magnitude smaller compared to the differences in the integral thrust. For this reason a classical grid convergence study (GCI) can be sometimes misleading, since local effects might be caught up by error compensation. However, these local effects are particularly important the present case where a detailed analyses of three-dimensional features are studied. Probably, a "bad" grid in the root region with for example large skew angles, aspect ratios or under-resolved boundary layers would not allow allow for a detailed evaluation of the relevant flow features, but on the other hand would also not be reflected in a global GCI.

Although, the general impact of the grid on the solution seems to be very small, it must be noted that for the lower wind speeds the local effects of the aerodynamic modifications on the overall blade performance can come into the same order of magnitude as the accuracy of the CFD framework. This fact is analyzed in the newly introduced section 4.6.

Regarding the grid dependency analysis, the paper has been modified in $\boxed{\text{R2:G2-a}}$ (page 7, line 181) and $\boxed{\text{R2:G2-b}}$ (page 10, line 224).

3. "*Are there any manufacturing constraints for the proposed solutions to increase aerodynamic efficiency?*"

This study is based on results of the research project AssiSt which was a cooperation with industry. The purpose was to get a better understanding of the basic nacelle parameters and the root aerodynamics of the root region of a wind turbine being equipped with flatback airfoils. At first instance the study neglected any other engineering disciplines. Therefore, no strict geometric constraints were imposed. Some considerations, like the lateral blade shifting, would impose a fundamental re-engineering of the engine construction.

Other modifications, primarily the blade root fairing, would easily be applicable from an engineering point of view since the blade connectors are fixed to the spinner. However, this leaves aside any economic deliberation.

An overview on the project results can be found in

Kühn, Timo, et al. "Results of the research project AssiSt." Journal of Physics: Conference Series. Vol. 1037. No. 2. IOP Publishing, 2018.

4. "*The proposed nacelle modifications and their impacts locally, on the root flow have been shown in detail. However, it would be interesting to quantify the impact of these solutions on a global parameter performance, such as the total power produced of the turbine*"

This comment is off course very important, since the crucial parameter that counts is the amount of energy extracted from the wind! Whenever it was possible, the relative improvements or degradations of the rotor efficiency have been added in the text for each of the analyzed modifications:

- regarding the difference of the baseline geometry to the isolated rotor $\boxed{\text{R2:G4-a}}$ (page 18, line 385)

- regarding the impact of the nacelle thickness $\boxed{\text{R2:G4-b}}$ (page 22, line 454)

- regarding the relative movement of the blade relative to the nacelle $\boxed{\text{R2:G4-c}}$ (page 28, line 540)

- In section 4.6 the global assessment of the fairing type modification has been analyzed. Additional wind speeds have been considered that mimic off-design conditions. While the benefits obtained at low wind speeds are very small, they are considerable at higher wind speeds, where additionally a favorable effect can be expected for future investigations that take into account high levels of atmospheric turbulence. **R2:G4-d** (page 33, line 602) and **R2:G4-e** (page 38, line 684)

**List of the major changes in the manuscript**

The line numbers correspond to the marked-up manuscript, not ot the revised version of the manuscript.

- Affiliation: Removed Typo in the address of the authors Kühn and Altmikus
- Abstract: small changes in the wording
- Introduction:
  - Added the subsection 1.3 (page3, line 66) focusing on literature dealing with the interacting flow fields of the rotor and the nacelle
  - Added the subsection 1.4 (page3, line 81) focusing on literature dealing with possible technologies that can be utilized to improve the aerodynamic efficiency in the root region.
  - In subsection 1.5 (page4, line 97) a clearer distinction of the scope of work is made with respect to section 1.4
- In section 2 a more detailed description of the reference turbine is included (page4, line 109)
- Computational details
  - Further references are given that shall prove the validity of the employed CFD solver and framework. (page 5 line 144)
  - The effect of boundary layer transition on the aerodynamics of the root region is discussed (page 5 line 154 – page 6 line 167)
- Results
  - The influence of the numerical grid on the solution is presented in the new subsection 4.1 (page 10, line 224-236).
  - The effect of the geometrical modifications on the overall performance has been added and edited in (page 18, line 385-389), (page 22, line 454-460), (page 28, line 540-548), (page 34, line 625-644)
  - An assessment of other wind speeds (8, 12, 15m/s) apart from the design wind speed is included in section 4.6 (page 33 from line 601)
- Conclusions
  - A summary of the results in off-design conditions as well as a characterization of the overall benefit of the considered modifications is included (page 38, line 684-692)
- Appendix B
  - A comparison between fully turbulent and transitional simulations is presented (page 38 line 711-739).

[revised manuscript text omitted]